# Early-life thymectomy leads to an increase of granzyme-producing γδ T cells in children with congenital heart disease

Alexa Cramer[1,8], Tao Yang [1,8], Lennart Riemann[1,2], Vicente Almeida [1], Christoph Kammeyer[3], Yusuf E. Abu [1], Elisa Gluschke[3], Svea Kleiner[3], Ximena León-Lara[1], Anika Janssen[1], Alejandro Hofmann[4], Alexander Horke[5], Constantin von Kaisenberg[6], Reinhold Förster [1,7], Philipp Beerbaum[3], Martin Boehne [3,9] & Sarina Ravens [1,7,9] ✉

Congenital heart disease (CHD) is the most common birth defect in newborns, often requiring cardiac surgery with concomitant thymectomy that is known to increase disease susceptibility later in life. Studies of γδ T cells, which are one of the dominant T cells in the early fetal human thymus, are rare. Here, we provide a comprehensive analysis of the γδ T cell compartment via flow cytometry and next-generation sequencing in children and infants with CHD, who underwent cardiac surgery shortly after birth. A perturbation of the γδ T cell repertoire is evident, and Vδ1 T cell numbers are reduced. However, those cells that are present, do retain cytotoxicity. In contrast, GZMA+CD28+CD161hi innate effector Vγ9Vδ2 T cells are found in higher proportions. TCR-seq identifies an increase in *TRDJ3*+ γδ T cell clones in children with CHD, but not in a confirmatory group of neonates prior to CHD surgery, which overall points to a persistence of fetal-derived effector γδ T cells in children with CHD.

Congenital heart disease (CHD) encompasses a spectrum of inborn heart defects that are the most common congenital anomalies worldwide, affecting approximately 1 in 100 newborns[1,2]. The majority of children with CHD require surgical intervention in early life[2]. Mostly due to vastly improved diagnosis and treatment options, CHD-related infant mortality has decreased by around 40% in the last 30 years[3]. Depending on the cardiac defect, now 97% of children with CHD reach adulthood, and most have a near-normal life expectancy, rendering research on the long-term implications of CHD treatments highly relevant[4,5].

During corrective or palliative congenital heart surgery, the complete or partial removal of the thymus–the organ for T cell development–is necessary in order to enable surgical access to the heart and great vessels[6]. In fact, the prevalence of thymectomy as a consequence of the surgical treatment for CHD has risen over the years alongside higher survival rates[7]. However, large population studies have suggested an association between thymectomy and an increased risk of all-cause mortality, autoimmune diseases, infections and cancer later in life[8–10]. Underlying perturbations of the immune system following thymus tissue removal may contribute to this increased

[1]Institute of Immunology, Hannover Medical School, Hannover, Germany. [2]Department of Pediatric Pneumology, Allergology and Neonatology, Hannover Medical School, Hannover, Germany. [3]Department of Pediatric Cardiology and Intensive Care Medicine, Hannover Medical School, Hannover, Germany. [4]Department of Pediatric Surgery, Hannover Medical School, Hannover, Germany. [5]Department of Cardiothoracic, Transplantation and Vascular Surgery, Hannover Medical School, Hannover, Germany. [6]Department of Obstetrics, Gynecology and Reproductive Medicine, Hannover Medical School (MHH), Hannover, Germany. [7]Cluster of Excellence RESIST (EXC 2155), Hannover Medical School, 30625 Hannover, Germany. [8]These authors contributed equally: Alexa Cramer, Tao Yang. [9]These authors jointly supervised the work: Martin Boehne, Sarina Ravens. ✉e-mail: ravens.sarina@mh-hannover.de

susceptibility to infection and malignancy[11–13]. In fact, previous studies reported that reduced thymic activity leads to changes in the immune cell composition in patients with CHD, including lower total numbers of T cells[14–16], lower frequencies of naïve CD4+ and CD8+ αβ T cells[17], and less diversity in the T cell receptor (TCR) repertoire of αβ T cells[18]. However, T cells that predominantly develop in the prenatal thymus[19], have not been the focus of studies in this context.

The development and function of T cell subsets are largely attributed to age, which dictates the origin of T cell precursor cells, the thymic microenvironment and thymic activity[20–22]. T cells that develop in the early-life thymus were described to differ in their expressed TCRs, longevity and responsiveness from postnatal-derived conventional T cells[22]. Specifically, one subset of T cells, namely γδ T cells, develop in early life[19]. γδ T cells are defined by their γδ T cell receptor (TCR), which is generated during their thymic development by somatic V(D)J gene recombination, accompanied by insertion of nucleotides (N) at the gene junctions, exponentially augmenting the overall repertoire diversity. In the fetal thymus, γδ T cells appear among the first lymphoid cells, start to develop around post-conceptional week 7 to 9 and are subsequently released in developmental waves from the thymus[19]. Recent advances in next-generation sequencing (NGS) approaches identified γδTCR features, such as low number of N-insertion and defined V(D)J gene element usage that clearly connect to a fetal-thymic origin[23], rendering these a valuable tool to define the developmental origin of human γδ T cells.

Accordingly, human γδ T cells are classified into γδ T cells that express (semi)invariant Vγ9Vδ2+ TCR and such that are characterized by a highly diverse Vδ1+ TCR repertoire[19]. These two main subsets in humans further differ in their functionality, TCR responsiveness, and location in the body. The largest γδ T cell subset in blood usually represents the Vγ9Vδ2 T cells (also called Vδ2 T cells), which predominantly develop in the early fetal thymus. Still, some Vγ9Vδ2 T cells are also generated in the postnatal thymus[24]. The majority of fetal-derived Vγ9Vδ2 T cells acquires granzyme- and TNFα production capabilities during intra-thymic development to be highly reactive at birth[25,26]. In the fetal thymus and the perinatal period, these Vγ9Vδ2 T cells are biased toward GZMA-secretion, express CD161 and are CD28+CD27+ in neonates[27,28]. During the first year of life, age-dependent maturation traits of the γδ T cell compartment, e.g., reflected by a decrease of CD28+CD27+ and an increase of CD57+ and NKG2A+ Vγ9Vδ2 T cells, are evident[28]. The Vγ9Vδ2+ TCRs sense and respond on the dependence of butyrophilin-like (BTNL) molecules, small, nonpeptidic metabolites from the isoprenoid synthesis often referred as phosphoantigens (pAg). The by far most potent pAg for inducing polyclonal Vγ9Vδ2 T cell responses is the isopentenyl pyrophosphate and (E)−4-hydroxy-3-methyl-but-2-enyl pyrophosphate (HMBPP) compound, which derives from a variety of microbes[29,30]. Another γδ T cell subset found in the blood, albeit with individual profiles in adults, are the Vδ1 T cells[31]. Virus-reactive Vδ1 T cells have been described in congenital infections[32]. However, the majority of Vδ1 T cells present after birth develop as naïve cells in the postnatal thymus and differentiate into cytotoxic effector cells upon TCR recognition of host-cell-derived molecules, which are thought to be cellular stress signals in infections and malignant transformations[33].

Generally, γδ T cells have been shown to play a pivotal role in the neonatal immune system, when other adaptive immune cell populations are still less mature and not yet primed, providing immune surveillance and rapid defense responses against several pathogens such as cytomegalovirus[32] or bacterial sepsis[28]. Later on in the adolescent and adult immune system, γδ T cells are involved in the immune responses against a plethora of pathogens, inflammatory diseases and cancer responses[34]. In light of the importance of γδ T cells for the pediatric and adult immune system and their development in the pre- and postnatal thymus, the effects of thymectomy in the neonatal period on γδ T cells in children with CHD are of high scientific and clinical interest, but largely unknown[16,35].

In this study, we fill this gap by investigating the effects of thymectomy early after birth on γδ T cell phenotypes, TCR repertoires, gene expression profiles, and functionality in 5- to 12-year-old children with CHD and provide evidence for disruption of the γδ T cell compartment and an increase in cytotoxic effector γδ T cells after thymectomy. In addition, γδ T cells in neonates with CHD were assessed before thymectomy and 6 months later to define early-life maturation profiles in relation to thymic activity.

## Results

### Early-life thymectomy perturbs the αβ and γδ T cell compartment later in childhood

To understand the consequences of thymic tissue removal in the early postnatal period on peripheral blood T cells later in childhood, a study population of 5- to 12-year-old children with CHD who had undergone cardiac surgery early after birth (CHD) was established, and immune cells were investigated by flow cytometry, TCR-seq and scRNA-seq (Supplementary Table S1). First, peripheral blood immune cells were studied by flow cytometry in the children with CHD (n = 18) and compared with age-matched controls (n = 12, non-CHD) (Table 1). The children with CHD had complex and/or critical CHD requiring cardiac surgery with concomitant removal of thymic tissue (thymectomy) within the first 6 weeks of life (median of 8.5 days after birth) (Table 1).

To investigate postnatal thymic activity in the pediatric CHD group at 5–12 years after thymectomy, the CD31 expression on naïve CD4 T cells was examined[36,37]. Lower frequencies of CD31+ cells in naïve CD4 T cells in thymectomized children with CHD, but not in the control group, were indicative of a reduction in recent thymic emigrants (RTE) (Fig. 1a, Supplementary Fig. 1a). In line with this, significantly lower CD3 T cell frequencies within lymphocytes and pan-leukocytes were observed in children with CHD as compared to controls (Fig. 1b, Supplementary Fig. 1b), presumably resulting in higher monocyte and NK cell frequencies, but not B cell frequencies (Supplementary Fig. 1b). For αβ T cells, decreased CD4 and CD8 T cell frequencies within the lymphocyte compartment were evident in the children with CHD (Fig. 1c), whereas the composition of αβ T cell subtypes remained

**Table 1 | Summarized characteristics of children with CHD post-surgery and controls**

| Characteristic | Children with CHD (n = 18) | Ctrl (n = 12) |
|---|---|---|
| Age at sample collection | | |
| Median [IQR] | 10 years [6–11] | 6.5 years [5–9.75] |
| Age at the time point of cardiac surgery involving (partial) removal of thymic tissue | | |
| Median [IQR] | 8.5 days [5.5–13.75] | – |
| Female, no. (%) | 7 (39%) | 5 (42%) |

CHD congenital heart disease, Ctrl control (non-CHD).

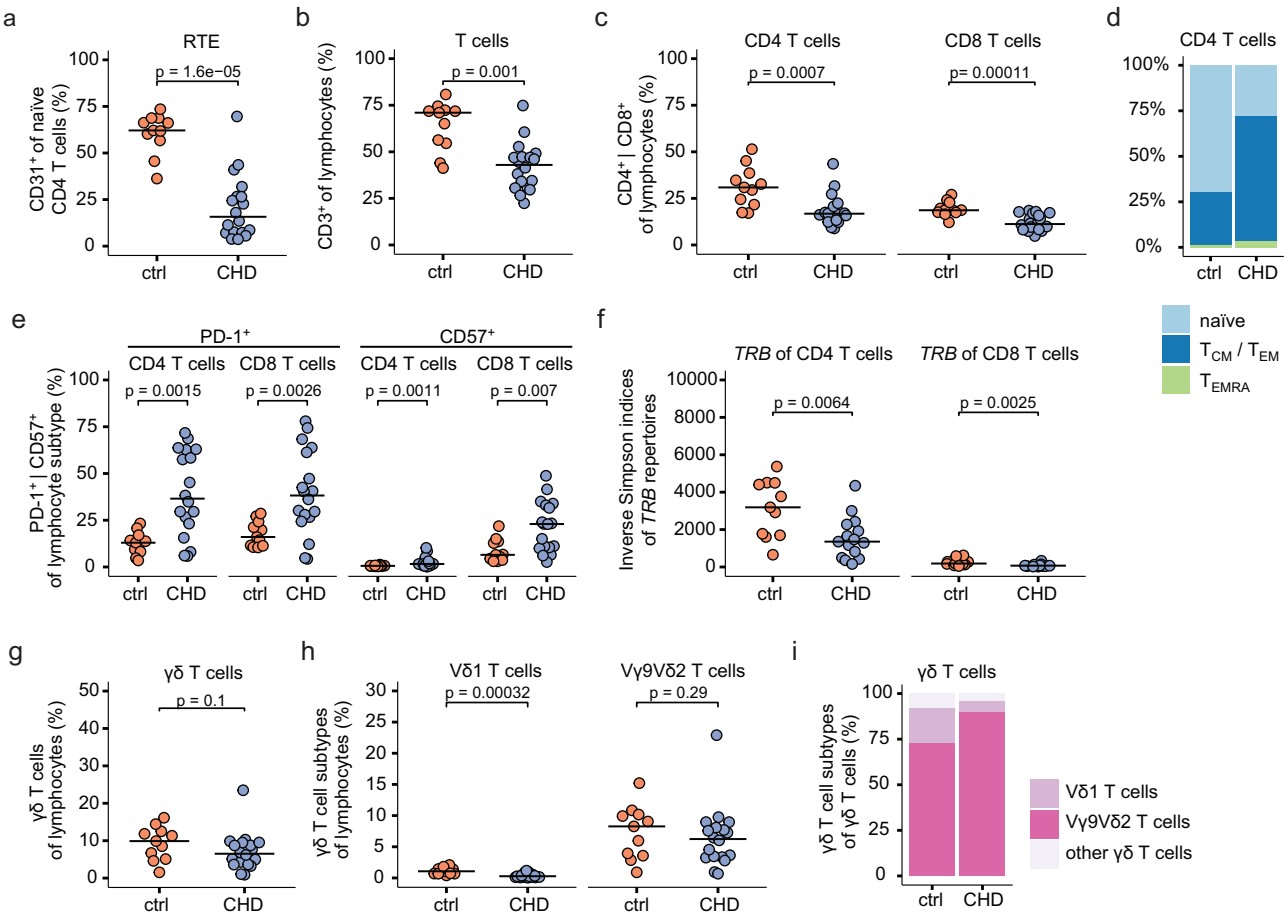

**Fig. 1 | Early-life thymectomy affects the αβ and γδ T cell populations.** Flow cytometric and TCR-seq analyses of peripheral mononuclear blood cells from 5- to 12-year-old children with CHD (n = 18, CHD) after cardiac surgery involving total or partial thymectomy, and age-matched non-CHD controls (n = 11, ctrl). **a** Percentage of CD31⁺ cells within naïve CD4 T cells, indicative of Recent Thymic Emigrants (RTE). **b** Dot plot shows CD3 T cell frequency of lymphocytes (defined as FSC-A^low/SSC-A^low). **c** Percentage of CD4 T cells (left) and CD8 T cells (right) among lymphocytes. **d** Stacked bar plot showing the percentage of a naïve (CD45RA⁺CCR7⁺), memory (CD45RA⁻, either CCR7⁺ ($T_{CM}$) or CCR7⁻ ($T_{EM}$)) or terminally differentiated effector memory phenotype ($T_{EMRA}$; CD45RA⁺CCR7⁻) among CD4 T cells. **e** Dot plot showing the percentage of PD-1 (left) or CD57 (right) expression among CD4 T cells

and CD8 T cells, respectively. **f** *TRB* repertoire analysis of FACS-sorted CD4 or CD8 T cells. Inverse Simpson diversity indices of CD4 T cells (left) and CD8 T cells (right). **g** Dot plots present the percentage of CD3⁺ γδ T cells among lymphocytes. **h** Percentage of CD3⁺γδTCR⁺ Vδ1 or CD3⁺γδTCR⁺ Vγ9Vδ2 T cells within lymphocytes. **i** Stacked bar plots present the abundance of respective γδ T cell subsets as median values within ctrl and children with CHD. Statistical analyses were performed using the two-sided Wilcoxon–Mann–Whitney U test; not significant (ns). No adjustments for multiple comparisons were made. Horizontal bars indicate median values. Each dot represents one donor. Source data are provided as a Source Data file.

stable among CD3 T cells (Supplementary Fig. 1c). Flow cytometric analyses of surface markers, namely CD45RA and CCR7, which define naïve and memory T cells, showed lower frequencies of naïve CD4 and CD8 T cells (CD45RA⁺CCR7⁺) and an increase in memory T cells in the children with CHD as compared to their age-matched counterparts (Fig. 1d, Supplementary Fig. 1d, e), which is consistent with previous reports[17,18]. PD-1 expression on both CD4 and CD8 T cells was higher in children with CHD (Fig. 1e, Supplementary Fig. 1f). Moreover, they showed elevated levels of CD57 expression (Fig. 1e), a marker associated with a senescent-like cell state[38], in the thymectomized children with CHD. Next, PBMC samples were stimulated with anti-CD3 and anti-CD28 for 48 h, followed by the analysis of CD4 and CD8 T cells. There were no major differences in GZMB and TNFα production, and the cellular activation markers HLA-DR and CD69 among controls and children with CHD (Supplementary Fig. 1g). In addition, the diversity of the T cell receptor ß-chain (*TRB*) repertoire was assessed by next-generation sequencing (NGS) of FACS-sorted CD4 and CD8 T cells. For both subsets, a lower *TRB* repertoire diversity as measured by the inverse Simpson index was confirmed[18] (Fig. 1f). The number of nucleotide insertions within the variable V(D)J regions was marginally

lower in the children with CHD compared to the non-CHD controls, exclusively for the *TRB* repertoire of CD4 T cells, but not CD8 T cells (Supplementary Fig. 1h).

Next, the abundance of γδ T cells was examined. In contrast to CD4 and CD8 T cells, there was only a slight reduction of γδ T cells among the lymphocytes in the children with CHD who underwent thymectomy as neonates (Fig. 1g). The expressed γδ TCR can classify human γδ T cells into Vγ9Vδ2 T cells and Vδ1 T cells. The Vγ9Vδ2 T cells are one of the first major T cell subsets in the early fetal thymus[19], but can still be generated by the postnatal thymus[24]. Vδ1 T cells are the dominant γδ T cell population in the postnatal thymus[19]. Interestingly, Vδ1 T cells, but not Vγ9Vδ2 T cells, showed reduced frequencies within lymphocytes and total T cells in the CHD group (Fig. 1h, Supplementary Fig. 1i). Consistent with these results, the frequencies of Vγ9Vδ2 T cells within total γδ T cells were increased in the pediatric CHD group (Fig. 1i, Supplementary Fig. 1j).

In conclusion, children with CHD who have undergone cardiac surgery with concomitant thymectomy within the first 6 weeks of life show reduced numbers of αβ T cell subsets, reduced *TRB* repertoire diversity, enrichment for effector memory phenotypes and CD57/PD-1

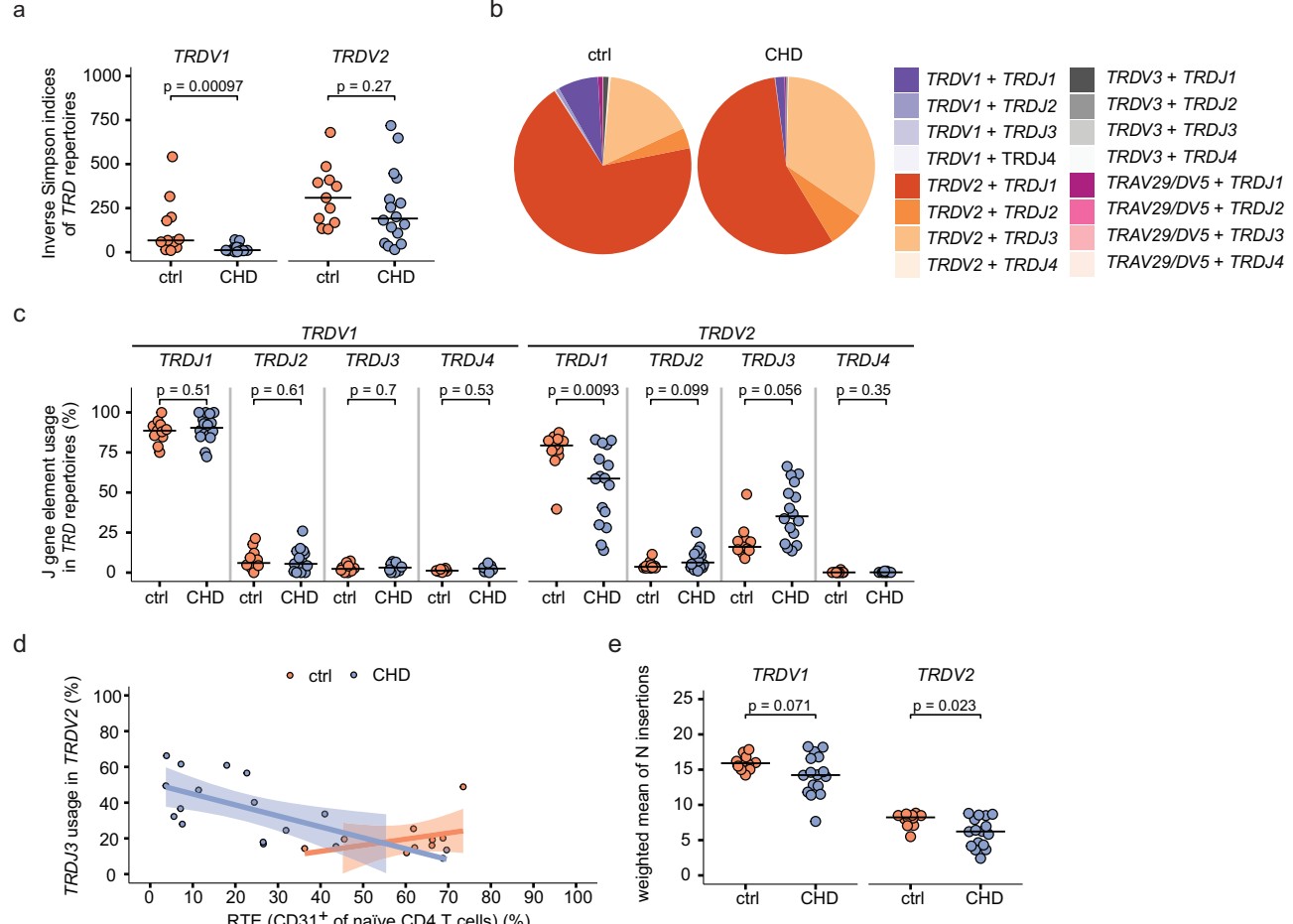

**Fig. 2 | Enrichment of *TRDV2*⁺ *TRDJ3*⁺ γδ T cell clones in children with CHD.** *TRD* repertoires analyses of FACS-sorted peripheral blood γδ T cells from 5- to 12-year-old children with CHD post-surgery (*n* = 16, CHD) and age-matched controls (*n* = 11, ctrl). **a** Inverse Simpson indices of *TRDV1*⁺ (left) and *TRDV2*⁺ (right) *TRD* repertoires. **b** Distribution of paired *TRDV* and *TRDJ* gene elements calculated from all ctrl and all CHD samples are depicted in the pie chart. The size of pie segments represents the median value of respective gene pairs calculated from all ctrl or CHD samples. **c** J-gene usage in *TRDV1*⁺ (left) and *TRDV2*⁺ (right) *TRD* sequences in percent, differentiated according to respective J-gene (*TRDJ1*, *TRDJ2*, *TRDJ3*, *TRDJ4*). **d** Percentage of RTE (CD31⁺ cells of naïve CD4⁺ T cells) plotted against the

proportion of *TRDJ3*⁺*TRDV2*⁺ sequences for children with CHD, showing negative correlation (Spearman's rho = −0.69, *p*-value = 0.004), and the age-matched controls (Spearman's rho = 0.36, *p*-value = 0.33). The error band describes the standard deviation of the displayed data points. **e** N-insertion counts in *TRDV1*⁺ (left) and *TRDV2*⁺ (right) sequences, calculated as weighted mean, weighted by the total number of clones per sample. Statistical analyses were performed using the two-sided Wilcoxon–Mann–Whitney *U* test. No adjustments for multiple comparisons were made; not significant (ns). Bars indicate median values; each dot represents one donor. Source data are provided as a Source Data file.

expression on conventional T cells. The abundance of γδ T cells expressing a Vγ9Vδ2⁺ TCR suggests that a reduced postnatal thymic activity supports the postnatal persistence of γδ T cells, presumably developed prior to thymectomy.

### TCR-seq identifies enrichment of fetal-derived γδ T cell clones in CHD children that display reduced thymic output

Next, we aimed to determine the impact of thymectomy early after birth on the expressed TCR repertoire of γδ T cells in children with CHD. For this purpose, FACS-sorted peripheral blood γδ T cells from a subgroup of children with CHD (*n* = 16) and age-matched non-CHD controls (*n* = 11) were subjected to an mRNA-based NGS analysis of the γ-chain (*TRG*) and δ-chain (*TRD*) repertoires (Supplementary Fig. 2a)[39]. Initial analyses of the *TRG* repertoires depicted an enrichment of *TRGV9*⁺ clones in children with CHD (Supplementary Fig. 2b). In addition, a reduced *TRGV9* repertoire diversity as measured by the inverse Simpson index was observed in the 5- to 12-year-old children with CHD after surgery (Supplementary Fig. 2c). The latter was most likely due to a higher frequency of fetal-derived *TRGV9*⁺ T cell clones that have no N-insertions post-surgery[23].

For *TRD* repertoire analysis, we adapted the γδ T cell subset classification from the above FACS analysis and compared *TRDV1*⁺ (representing Vδ1 T cells) and *TRDV2*⁺ (mainly representing Vγ9Vδ2 T cells) *TRD* repertoires. Notably, we found a reduction in diversity for the *TRDV1*⁺ and *TRDV2*⁺ *TRD* repertoires in children with CHD after surgery, although this reduction was only significant for *TRDV1*⁺ sequences (Fig. 2a).

Fetal-derived γδ T cells can be identified by a low number of N-insertions and preferential use of *TRDJ3* gene elements within their *TRD* repertoire[23]. In contrast, postnatal-derived γδ T cell clones are known to have a higher TCR repertoire diversity and mainly use *TRDJ1*⁺ TRD clones[23,40]. Analysis of paired V-genes and J-genes indicates a higher prevalence of *TRDV2*-*TRDJ3* sequences in CHD children as compared to the controls, as depicted by combined and individual pie charts (Fig. 2b, Supplementary Fig. 2d). In this direction, an enrichment of *TRDJ3*⁺ sequences and a decrease of *TRDJ1*⁺ sequences among *TRDV2*⁺ clones, but not among *TRDV1*⁺ clones, was observed in the children with CHD as compared to the non-CHD controls (Fig. 2c). Importantly, *TRDJ3* gene element usage negatively correlates with the abundance of RTE as measured by the frequency of naïve CD31⁺ CD4

T cells in the children with CHD, while this is not the case in the control group (Fig. 2d). Vice versa, children with higher frequencies of RTE showed increased levels of *TRDJ1* gene elements (Supplementary Fig. 2e). Moreover, we observed lower N-insertion counts in the *TRDV1* and *TRDV2* sequences, the latter being statistically significant (Fig. 2e). Together, the increase in γδ T cells expressing a TCR with enrichment for early fetal thymus-derived *TRDJ3+* T cell clones suggests that a reduced postnatal thymic activity supports the persistence of γδ T cell clones that developed prior to surgery.

### Children with CHD show an increase in CD28^hi^CD161^hi^ Vγ9Vδ2 T cells

Next, we investigated the phenotypes of γδ T cells in thymectomized children with CHD and age-matched controls by systematic flow cytometric analysis. The antibody panel included lineage markers to identify Vγ9Vδ2 and Vδ1 T cells, and 18 phenotypic markers to define γδ T cell differentiation, such as CD27, CD28, CCR7, CD127 and CD45RA, cellular activation states such as CD57 and CD25, the checkpoint molecule PD-1, CD161 to define innate γδ T cells, and surface markers associated with cytotoxic γδ T cells such as CD16 and NKG2A[41–43]. Naïve γδ T cells are defined by the co-expression of CD27 and CD28[25,26]. We also examined the expression of CD4 and CD8α, as these have been reported to be detectable on γδ T cells (Supplementary Fig. 3a)[28]. Unsupervised clustering was performed, based on the expression of 18 surface markers of γδ T cells from the CHD ($n = 18$) and non-CHD controls ($n = 11$). We identified eight clusters (c1–c8) as projected in UMAP (Fig. 3a). None of the clusters were specific to the children with CHD or the non-CHD control group (Fig. 3a, Supplementary Fig. 3b). However, cluster c6 was predominantly composed of cells from the control group. Overlaying the TCR information for V-gene usage clearly distinguished the Vγ9Vδ2 T cell cluster (c1–c4), a cluster c5 that may represent Vδ3 T cells, and Vδ1 T cell clusters (c6–c8) (Fig. 3b, Supplementary Fig. 3c). Moreover, the eight clusters differed in the expression levels of the respective phenotypic markers (Fig. 3c, Supplementary Fig. 3d). The heatmap shows scaled expression values for each marker for better comparability, highlighting marker positivity (yellow) and negativity (violet), respectively (Fig. 3d).

For Vγ9Vδ2 T cells, the most abundant cluster c1 was composed of CD28^hi^ CD161^hi^ Vγ9Vδ2 T cells and may represent the most innate γδ T cell subset. The majority of these cells in cluster c1 expressed NKG2A, and none of them were positive for CD57. The other three CD161^low/neg^ Vγ9Vδ2 T cell clusters (c2–c4) differed in the expression of CD16, NKG2A and CD28, respectively (Fig. 3c, d, Supplementary Fig. 3e). For Vδ1 T cells, high CD31 expression was observed in clusters c6–c7, with c7 also being CD8+ and a small cluster c8 characterized as CD57^hi^ CD16+ (Fig. 3c, d, Supplementary Fig. 3e).

The frequency of each of the eight clusters (c1–c8) was then examined in the children with CHD and the non-CHD controls (Fig. 3e). Specifically, the Vγ9Vδ2 T cell cluster c1, defined as CD28^hi^ CD161^hi^, was significantly enriched in the CHD patient group as compared to the controls (Fig. 3e, f). For Vδ1 T cells, a reduction of the clusters c6–c7 with high CD31 expression was observed in the CHD patients (Fig. 3e, f). Consistent with these analyses, a non-significant increase in CD28+ CD161+ γδ T cells and a significant decrease in CD31+ Vδ1+ cells among γδ T cells was observed in children with CHD as compared to the control group (Fig. 3g, h).

In summary, early-life heart surgery for CHD with concomitant thymectomy supports the presence of CD28^hi^ CD161^hi^ Vγ9Vδ2 T cells in childhood and leads to a reduction of naïve CD31+ Vδ1 T cells.

### scRNA sequencing reveals enhanced cytotoxic gene expression in children with CHD who underwent thymectomy early after birth

Next, we investigated the transcriptional profiles of γδ T cells in children with CHD after thymectomy. To this end, scRNA-seq was performed on peripheral blood γδ T cells from 5- to 12-year-old children with CHD who received thymectomy early after birth (CHD, $n = 4$) and control children (ctrl, $n = 4$) (Table 2, Suppl. Table S1). Unsupervised clustering of 6364 control group and 8188 CHD group γδ T cells with adjustment for differential gene expression analysis and uniform manifold approximation and projection (UMAP) identified six clusters (c1–c6) (Fig. 4a). The heatmap illustrates the top 50 differentially expressed genes (DEGs) defining each cluster (Supplementary Fig. 4a). Based on the expression of TCR-delta chain transcripts (*TRDV1* and *TRDV2*), the clusters were grouped into a Vδ1+ cluster c1 and Vδ2+ clusters c2–c6 (Fig. 4b). The Vδ1+ cluster c1 and the Vδ2+ clusters c2–c3 were more abundant in the control group, and the Vδ2+ cluster c6 was highly abundant in children with CHD (Fig. 4c). Gene expression profiles of innate type 1 immunity cells, including expression of *KLRC1*, *KLRB1* or granzyme-encoding genes, were evident within Vδ2+ clusters c2–c6 (Fig. 4d). Overall, Vδ2 T cells divide into *GZMB+GZMK^low^* subsets represented by clusters c2 and c4, and *GZMB^neg^GZMK+* subsets presented by cluster c3 and clusters c5-c6. Still, clusters c2-3 co-expressed marker genes of naïve γδ T cells, such as *CCR7* and *CD27*, and show evidence of *PECAM1* (encoding CD31) expression (Fig. 4d).

For Vδ1 T cells, the cluster c1 consisted of naïve γδ T cells, indicated by expression of *KLF2*, *TCF7* or *CD27* (Fig. 4d). In addition, Vδ1 T cells (c1) showed the highest expression levels for *PECAM1* (Fig. 4d). Comparison of gene module scores for RTE in the clusters c1–c6, divided between control and CHD children, showed the highest expression of RTE genes in the Vδ1+ cluster c1 of the control group (Fig. 4e). In addition, there were no major differences in the ability to proliferate, as measured by a gene module score for proliferation, between cells from controls and children with CHD (Supplementary Fig. 4b).

To understand potential transcriptional differences between γδ T cells from control and CHD children, the expression of *TCF7*, a representative marker gene of naïve γδ T cells, was examined in the respective six clusters divided between the two groups (Fig. 4f). The data provide evidence for less naive Vδ1 T cells in children with CHD. Further analysis of gene module scores for cytotoxicity, divided between cells from controls and children with CHD, suggests an increased expression for genes encoding cytotoxic effector molecules (e.g., GZMA, GZMB, TNFα or IFNγ) of Vδ1 T cells (c1) in children with CHD (Fig. 4g). Importantly, examination of intracellular GZMB and GZMA production ex vivo by flow cytometry indicates increased GZMA and GZMB release from Vδ1 T cells in children with CHD compared to the control group (Fig. 4h, Supplementary Fig. 4c). Taken together, the transcriptional profiles within the γδ T cell compartment are perturbed in children with CHD who underwent thymectomy for congenital heart surgery early after birth. The data show that Vδ1 T cells are reduced in children with CHD and that the remaining cells have an increased cytotoxic capacity.

### Children with CHD show an increase of GZMA+ innate effector Vγ9Vδ2 T cells

Next, we aimed to understand whether Vγ9Vδ2 T cells retain their functionality in children with CHD. First, the transcriptional profiles of cluster c6 cells, a cluster that was predominantly present in CHD children, were investigated. DEG analysis of cluster c6 versus all other Vδ2 T cell clusters revealed higher levels of *CD28* and *CD8A* expression in cluster c6 (Fig. 5a). Notably, *CD8B* was also expressed in this cluster c6. Furthermore, cluster c6 was defined by transcripts associated with the cell cycle, such as *CDK6* and *TOP2B*, and anti-apoptosis genes *BCL11B* and *BCL2* (Supplementary Fig. 5a). Transcripts of potential IL-17 producers, namely *RORC* and *CCR6*, marker genes described for human tissue γδ T cells[44], including *CD69*, *EZR*, *TNFAIP3* and *ITGA4*, *ITGB1*, and *ZBTB16* (encoding PLZF) were also enriched in cluster c6, respectively (Supplementary Fig. 5b). Consistent with the increased frequencies of CD28^hi^CD161^hi^ Vγ9Vδ2 T cells in the CHD group

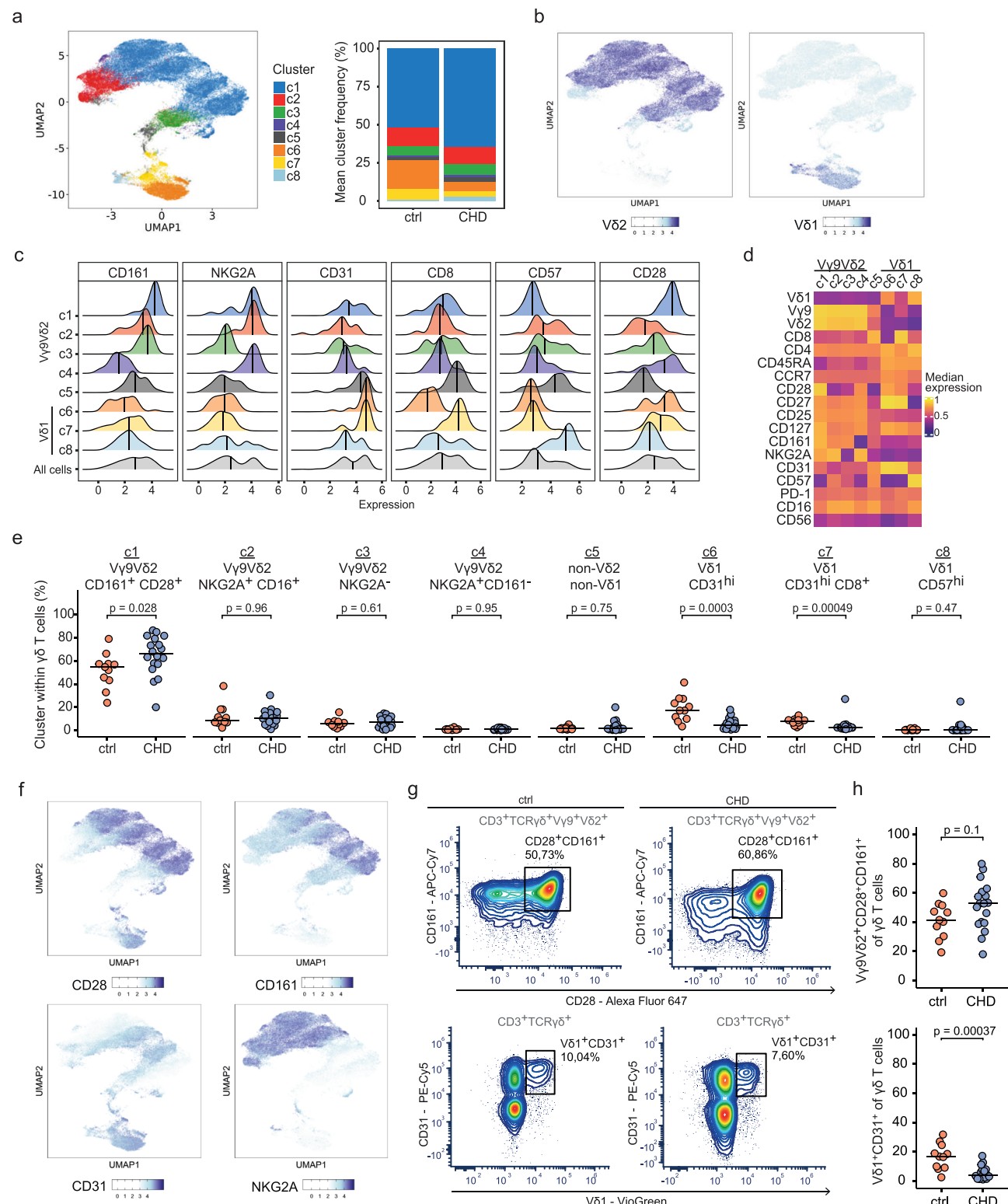

observed by FACS analysis (Fig. 3), expression of *KLRB1* (encoding CD161) and *CD28* was higher in Vδ2+ cluster c3, c5, and c6 cells of the CHD group as compared to controls (Fig. 5b).

Next, we examined the functionality of Vγ9Vδ2 T cells in children with CHD. The children with CHD showed enrichment of a specific Vδ2 effector cluster c6 (Fig. 4c), which was characterized by the expression profile *GZMA+GZMB−* (Fig. 4d) and high levels of *CD28* and *CD8A* (Fig. 5a). We therefore investigated the intracellular production of GZMA and GZMB by Vδ2 T cells, and how this relates to CD8A and

CD28 expression. As depicted in representative FACS dot plots, GMZA+ CD8A+CD28+ Vδ2 T cells were detectable in CHD children and age-matched healthy controls (Fig. 5c, Supplementary Fig. 5c). Most importantly, the frequencies of this Vδ2 T cell population were increased in the CHD group (Fig. 5d). In contrast, GZMB+ Vδ2 T cells were less frequent in the pediatric CHD group (Fig. 5e). However, despite the identification of *CD8B* transcripts in the cluster c6, the abundance of CD8B+ γδ T cells was not increased in children with CHD (Fig. 5f).

**Fig. 3 | Increase of CD28$^{hi}$CD161$^{hi}$ Vγ9Vδ2 T cells in children with CHD.** High-dimensional flow cytometry data analysis of children with thymectomy (CHD, $n = 18$) and non-CHD controls (ctrl, $n = 11$) reveals an increase CD28$^{hi}$CD161$^{hi}$ Vγ9Vδ2 T cells in 5- to 12-year-old CHD children after early-life cardiac surgery. **a** UMAP of obtained clusters (c1–c8) from pre-gated live CD3$^+$TCRγδ$^+$ cells, clustered by an unsupervised clustering approach based on surface marker expressions, and respective cluster proportions per group presented as stacked bar plot. **b** Expression UMAPs of obtained clusters colored by Vδ2 and Vδ1 surface marker expression. **c** Ridge plots show the expression of selected markers (CD161, NKG2A, CD31, CD8, CD57 and CD28) in each cluster. The vertical lines indicate the median marker expression within the respective cluster. **d** Heatmap visualizing the median scaled expression of surface markers within each cluster c1–c8. **e** Cluster frequencies among γδ T cells, compared between pediatric CHD patients and non-CHD controls. The clusters c1–c8 were annotated based on their surface marker expression characteristics. **f** Expression UMAPs of obtained clusters colored by surface marker expression of cluster defining markers. **g** Manual gating of Vγ9Vδ2$^+$CD28$^+$CD161$^+$ and Vδ1$^+$CD31$^+$ γδ T cell populations in representative samples. **h** Percentage of Vγ9Vδ2$^+$CD28$^+$CD161$^+$ and Vδ1$^+$CD31$^+$ cells among all γδ T cells. Statistical analyses were performed using the two-sided Wilcoxon–Mann–Whitney $U$ test; not significant (ns). No adjustments for multiple comparisons were made. Bars indicate median values, with each dot representing a single donor. Source data are provided as a Source Data file.

**Table 2 | Summarized characteristics of children with CHD post-surgery used for single-cell RNA sequencing**

| Characteristic | Children with CHD ($n = 4$) | Ctrl ($n = 4$) |
|---|---|---|
| Age at sample collection | | |
| Median [IQR] | 7 years [6.7–8.75] | 7.5 years [4.75–10.5] |
| Age at the time point of cardiac surgery involving (partial) removal of thymic tissue | | |
| Median [IQR] | 8 days [6.75–10.25] | – |
| Female, no. (%) | 0 (0%) | 2 (50%) |

In a second set of experiments, peripheral blood samples from another subgroup of children with CHD and non-CHD controls were subjected to in vitro stimulation with IL-2 and HMBPP, the latter known to be a potent activator of the Vγ9Vδ2$^+$ TCR[29,30,45,46], or IL-2 alone. Effector functions, namely intracellular GZMB and TNFα expression, of Vγ9Vδ2 T cells were examined at day 7 post-stimulation. The representative 2D plot from a child with CHD shows GZMB production in both in vitro settings (Fig. 5g), confirming the intrinsic ability of Vγ9Vδ2 T cells to express GZMB[47]. Despite that, slightly reduced levels of GZMB expression were evident in the children with CHD following HMBPP stimulation (Fig. 5g), and this is consistent with the decrease in GZMB$^+$ cells in the CHD group that were directly analyzed ex vivo within an independent experiment (Fig. 5d). After HMBPP stimulation, TNFα production increased to approximately 15% of Vγ9Vδ2 T cells in both groups (Fig. 5h, Supplementary Fig. 5d). In addition, no differences in surface marker expression of HLA-DR, CD69, PD-1 and CD57 of 7-day HMBPP-stimulated Vγ9Vδ2 T cells were evident among both groups (Supplementary Fig. 5e).

In conclusion, the transcriptional profiles of Vγ9Vδ2 T cells are different in children with CHD who underwent thymectomy early in life compared to age-matched children. Nevertheless, the responsiveness and functionality of granzyme-producing Vγ9Vδ2 T cells were preserved and shifted toward an increase in GZMA$^+$ effectors in the children with CHD.

### γδTCR repertoires are not perturbed before heart surgery in neonates with CHD

Next, we asked whether the perturbation of the γδ T cell compartment in the children with CHD was a result of reduced postnatal thymic activity due to cardiac surgery with thymic tissue removal performed, or a prenatal developmental disruption due to the CHD. For this reason, a second study cohort of 2- to 15-day-old newborns ($n = 15$; median age in days = 8) with CHD was established. Similar to the pediatric CHD group described above, these neonates had complex or critical congenital heart defects requiring heart surgery in the neonatal period (Table 3, Supplementary Table S2). After FACS sorting of γδ T cells from peripheral blood samples collected prior to congenital heart surgery, NGS of the TRG and TRD repertoires was performed. γδ TCR repertoires from 4–14 days old preterm neonates ($n = 10$)[48] and 2- to 3-day old healthy term neonates ($n = 6$), which together had a median age of 5 days, served as the non-CHD control group (Table 3).

There were no statistically significant differences in either the repertoire diversity or in the frequency of $TRGV9^+$ sequences within the $TRG$ repertoire of neonates with CHD and control neonates (Fig. 6a, b). In addition, the inverse Simpson diversity index for $TRDV2^+$ clones was similar in both groups (Fig. 6c). The pie charts suggest a $TRDJ3$-enriched $TRD$ repertoire in this age group that was independent of the CHD (Fig. 6d, Suppl. Fig. 6a). Next, we quantified the J-gene element usage within $TRDV2^+$ clones, and found that $TRDJ3$ was the most frequently usedgene segment to be used in both CHD and non-CHD neonates (Fig. 6e). Furthermore, another characteristic of the fetal TCR repertoire, namely a low count of N-insertions, was observed in all neonates regardless of the presence of CHD (Fig. 6f).

In conclusion, the γδTCR-seq results indicate that CHD per se does not affect γδTCR repertoires in neonates. It provides proof of concept that a reduced thymic activity during childhood leads to the persistence of fetal thymus-derived Vγ9Vδ2 T cell clones after surgery, resulting in preferential abundance of these cells in the 5- to 12-year-old children with CHD.

### Evidence for an increased proliferative capacity of γδ T cells in infants with CHD early after thymectomy

Lastly, we aimed to elucidate the impact of congenital heart surgery with concomitant removal of thymic tissue on the postnatal maturation of γδ T cells within the first 6 months of life. Therefore, we performed scRNA-seq of isolated peripheral blood γδ T cells from two longitudinally followed infants with CHD before (Baseline, BSL, on day 8 and 9 of life) and 6 months after surgery (Follow-up, FU, at day 188 and 189 of life) (Supplementary Table S2). Transcriptional profiles of peripheral blood γδ T cells from the two longitudinally followed age-matched, uninfected preterm neonates without CHD (Ctrl at BSL and FU, GSE245131[28]) were used as controls. Transcripts from 2962 CHD baseline (CHD BSL), 2914 CHD follow-up (CHD FU), 5356 non-CHD baseline (Ctrl BSL), and 6669 non-CHD follow-up (Ctrl FU) γδ T cells were obtained and classified into 8 clusters (Fig. 7a). The DEGs defining each cluster are presented in the heatmap (Supplementary Fig. 7a). Based on the expression of TCR-delta chain genes, the clusters are divided into Vδ1$^+$ clusters c1–c3 and Vδ2$^+$ clusters c4-c8 (Supplementary Fig. 7b).

Analysis of the gene module scores for recent thymic emigrants (RTEs) per cluster and time point indicated decreased thymic activity after surgery in the CHD group (Supplementary Fig. 7c). Specifically,

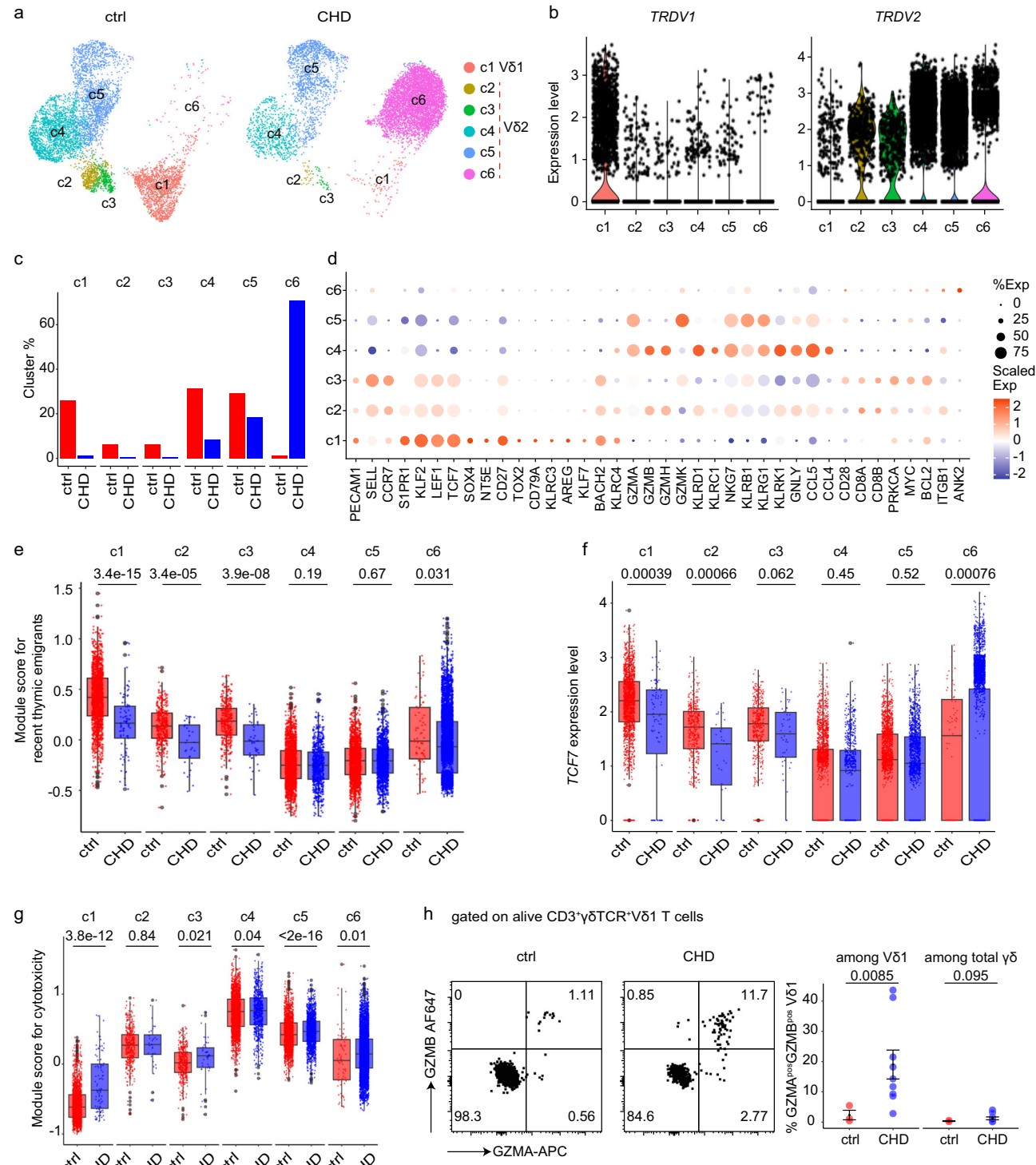

**Fig. 4 | Elevated expression of cytotoxicity gene module scores in thymecto-mized children with CHD. a** UMAP visualization of identified clusters from scRNA-seq of FACS-sorted γδ T cells from 5- to 12-year-old children who received thy-mectomy early after birth (CHD, *n* = 4) and control children (ctrl, *n* = 4), colored by cluster. **b** The expression of *TRDV1* and *TRDV2* transcripts per each cluster. **c** The bar plot displays fractions (%) of absolute cell numbers from ctrl and CHD γδ T cells that contributed to clusters c1–c6. **d** Dot plot of the average gene expression (columns) per cluster (rows). Dots are colored by average logFC (Scaled Exp) and sized by the percentage of cells per cluster that expressed this gene (% Exp). **e**–**g** Box plots of the single-cell gene signature module score for recent thymic emigrants (RTE) (**e**), *TCF7* (**f**), and cytotoxicity (**g**) of each cluster in both conditions. The RTE module score was computed based on *KLF2, CCR9, PECAM1, S1PR1, LEF1, TCF7, SOX4, NT5E*, and *SELL*. The cytotoxicity score was computed based on *GZMA,*

*GZMB, GZMH, GZMK, NKG7, GNLY, KLRK1, KLRD1, KLRF1, KLRC1, KLRB1, KLRG1, CCL5, CCL4*, and *CXCR6*. **h** Left: protein expression of GZMA and GZMB in one representative ctrl and one CHD child on alive CD3⁺γδTCR⁺ Vδ1 T cells. Right: the frequency of GZMA^pos^GZMB^pos^ cells among Vδ1 T cells and total γδ T cells shown as mean ± SEM. $n_{ctrl}$ = 3, $n_{CHD}$ = 9. **e**–**g** The cell numbers of each cluster from 4 ctrls and 4 CHDs are as follows: c1, nctrl = 1653, nCHD = 91; c2, nctrl = 405, nCHD = 32; c3, nctrl = 389, nCHD = 42; c4, nctrl = 1989, nCHD = 696; c5, nctrl = 1851, nCHD = 1511; c6, nctrl = 77, nCHD = 5816. The minima of boxplot: the lowest data point within 1.5 times the first quartile's Interquartile Range (IQR). Maxima: the highest data point within 1.5 times the IQR of the third quartile. Center: median of the data. **e**–**h** Statistical significance was determined by unpaired two-sided Wilcoxon–Mann–Whitney *U* test; not significant (ns). Source data are provided as a Source Data file.

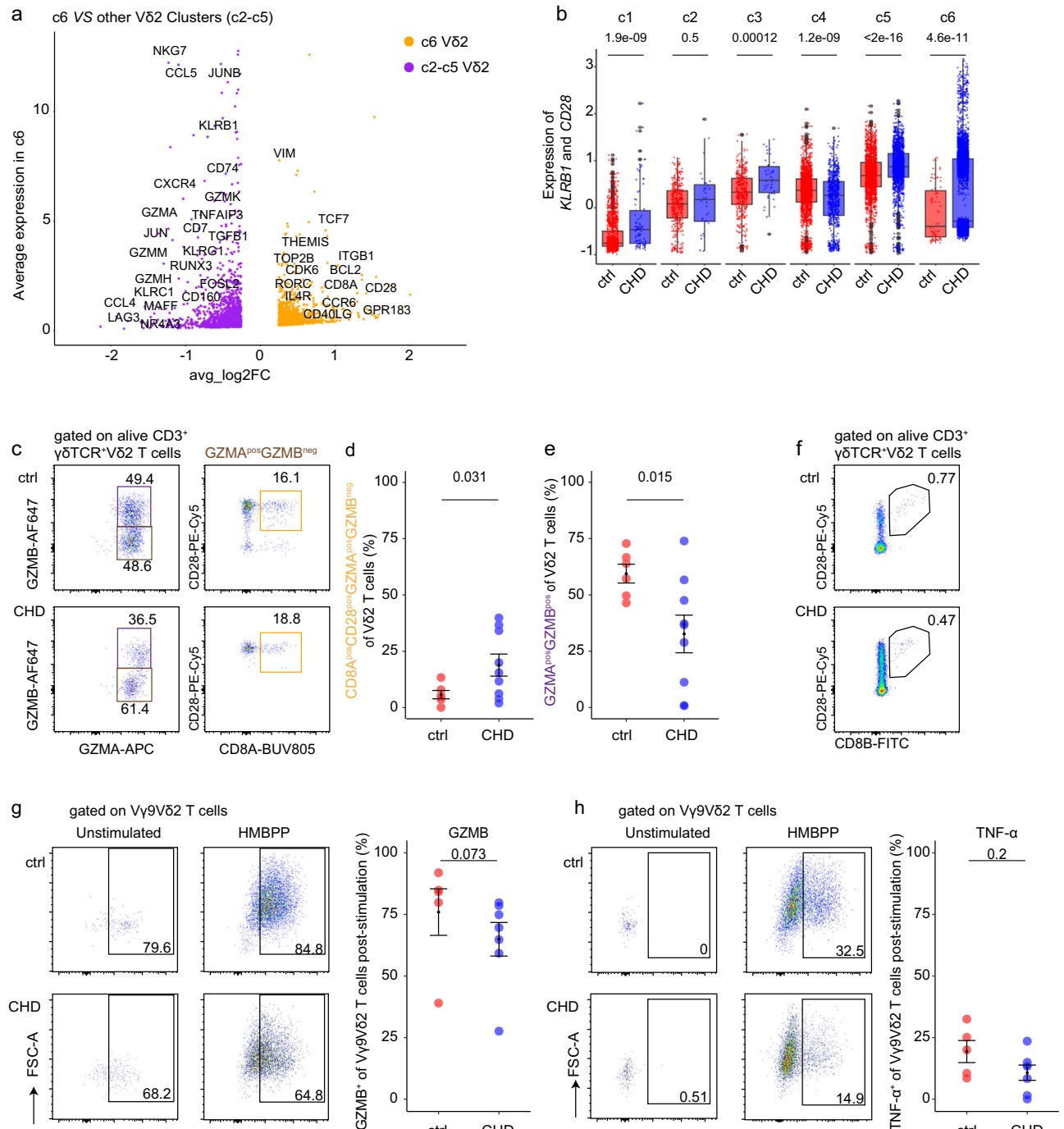

**Fig. 5 | Increase of GZMA+ innate effector Vδ2 T cells in children with CHD.**
**a** Volcano plot shows the average expression of the differentially expressed genes between Vδ2 cluster c6 and other Vδ2 clusters (c2-c5). **b** Box plots of the single-cell gene signature module score for *KLRB1* and *CD28*. The cell numbers of each cluster from 4 ctrls and 4 CHDs are as follows: c1, $n_{ctrl}$ = 1653, $n_{CHD}$ = 91; c2, $n_{ctrl}$ = 405, $n_{CHD}$ = 32; c3, $n_{ctrl}$ = 389, $n_{CHD}$ = 42; c4, $n_{ctrl}$ = 1989, $n_{CHD}$ = 696; c5, $n_{ctrl}$ = 1851, $n_{CHD}$ = 1511; c6, $n_{ctrl}$ = 77, $n_{CHD}$ = 5816. The minima of boxplot: the lowest data point within 1.5 times the first quartile's Interquartile Range (IQR). Maxima: the highest data point within 1.5 times the IQR of the third quartile. Center: median of the data. **c** Flow cytometric analyses depict GZMA, GZMB, CD28 and CD8A expression of unstimulated alive CD3+γδTCR+Vδ2+ T cells in one representative control and one child with CHD. **d, e** The frequency of CD8A$^{pos}$CD28$^{pos}$GZMA$^{pos}$GZMB$^{neg}$ (**d**) and GZMB$^{pos}$ (**e**) among Vδ2 T cells, shown as

mean ± SEM. $n_{ctrl}$ = 6, $n_{CHD}$ = 9. **f** Representative FACS plot of CD28 and CD8B expression on Vδ2 T cells in one control (1 out of 1) and one child with CHD (1 out of 4). **g, h** In vitro HMBPP + IL-2 stimulation of peripheral blood samples of 5- to 12-year-old CHD children (*n* = 7) and controls (*n* = 5), or IL-2 alone was performed. **g** Intracellular GZMB expression in both unstimulated (only IL-2) and stimulated (IL-2 plus HMBPP) PBMCs of alive/CD3+/γδTCR+/Vγ9Vδ2 T cells in representative samples, and as determined by frequency at day 7 post-stimulation. **h** TNFα-producing Vγ9Vδ2 T cells in one representative ctrl and CHD sample, and determined by frequency at 7 days post-stimulation. **c–g** Statistical analyses were performed using by unpaired two-sided Wilcoxon–Mann–Whitney *U* test; not significant (ns). Bars indicate median values; each dot is representative of one donor. The error band describes the standard deviation of the displayed data points. Source data are provided as a Source Data file.

genes related to RTE were less expressed in the Vδ1+ clusters c2–c3, and the Vδ2+ clusters c6–7 (Supplementary Fig. 7c).

Next, the eight clusters were annotated to transcriptional profiles of naïve and effector γδ T cell subsets. For this, genes expressed by naïve T cells (e.g., *CCR7* or *TCF7*) and granzyme-encoding genes, natural killer receptors, genes expressed by various γδ T cell subsets, including *CCR6, RORC, CCR4* or *GATA3* were considered (Fig. 7b)[25,26,28]. For Vδ1 T cells, the clusters c1–c3, defined by *PECAM1, S1PR1, KLF3* and *TCF7* expression, were identified. For Vδ2 T cells, cluster c4 represents *CCR4*+ γδ T cells, and cluster c5 cells express *CCR6* and *RORC*. Cytotoxic effector γδ T cells were evident in the Vδ2+ clusters c6–c7, dividing into GZMB^hi cells in c7 and GZMB^low cells in c6 (Fig. 7a, b). Next, the quantification of the respective clusters in controls and CHD infants with

respect to time points, namely before surgery (BL) and at 6 months later (FU), shows a reduction of Vδ1+ clusters and an enrichment of cytotoxic Vδ2+ effector cells (c6–c7), but not of type 2 (c4) and type3 (c5) immunity Vδ2+ clusters at 6 months after surgery (Fig. 7c, Supplementary Fig. 7d). Notably, differences in the abundance of the Vδ1+ and Vδ2+ T cell clusters at BL, namely a higher prevalence of Vδ2+ clusters c6–7 in the premature neonates forming the control group, could be explained by the prematurity state at birth (Fig. 7c).

Next, potential differences in transcriptional programs between control and CHD infants at the FU time point were investigated. DEG analysis among Vδ2+ cluster c6–7 cells of the control and CHD samples at the FU time point gives evidence for enhanced cytotoxic capabilities in CHD patients, marked by increased expression levels of genes encoding granzymes and various natural killer receptors (Fig. 7d). The expression of genes related to tissue T cells (e.g., *EZR, ITGA4, ITGZM,* and *VIM*) and proliferation (e.g., *BCL2, CDK6,* and *BCL2L1*) was also enriched in the CHD infants at 6 months after corrective heart surgery (Fig. 7d). However, the gene expression profiles of γδ T cells in the 5- to 12-year-old children with CHD, defined from the abundant cluster c6 (Fig. 4), were rarely detectable in the γδ T cells of the infants with CHD at the FU time point (Supplementary Fig. 7e).

Next, an enriched gene ontology analysis revealed that in addition to cell killing, transcripts associated with proliferation were enriched in the Vδ2+ clusters c6–7 (Fig. 7e). To support the idea of a homeostatic expansion of the respective γδ T cell subsets due to the reduced thymic activity early after surgery, gene module scores for proliferation were investigated with respect to cluster and sample group (Fig. 7f). Gene

### Table 3 | Summarized characteristics of neonates prior to surgery and neonatal controls

| Characteristic | Neonates with CHD (n = 15) | Neonatal ctrl (n = 16) |
|---|---|---|
| Age at sample collection | | |
| Median [IQR] | 8 days [5.5–10.5]* | 5 days [3–6.25] |
| Female, no. (%) | 2 (13%) | 7 (44%) |

*CHD* congenital heart disease, *Neonatal ctrl* neonatal control (non-CHD).

*For children with CHD, the age at sample collection is also the age at the time point of cardiac surgery involving (partial) removal of thymic tissue.

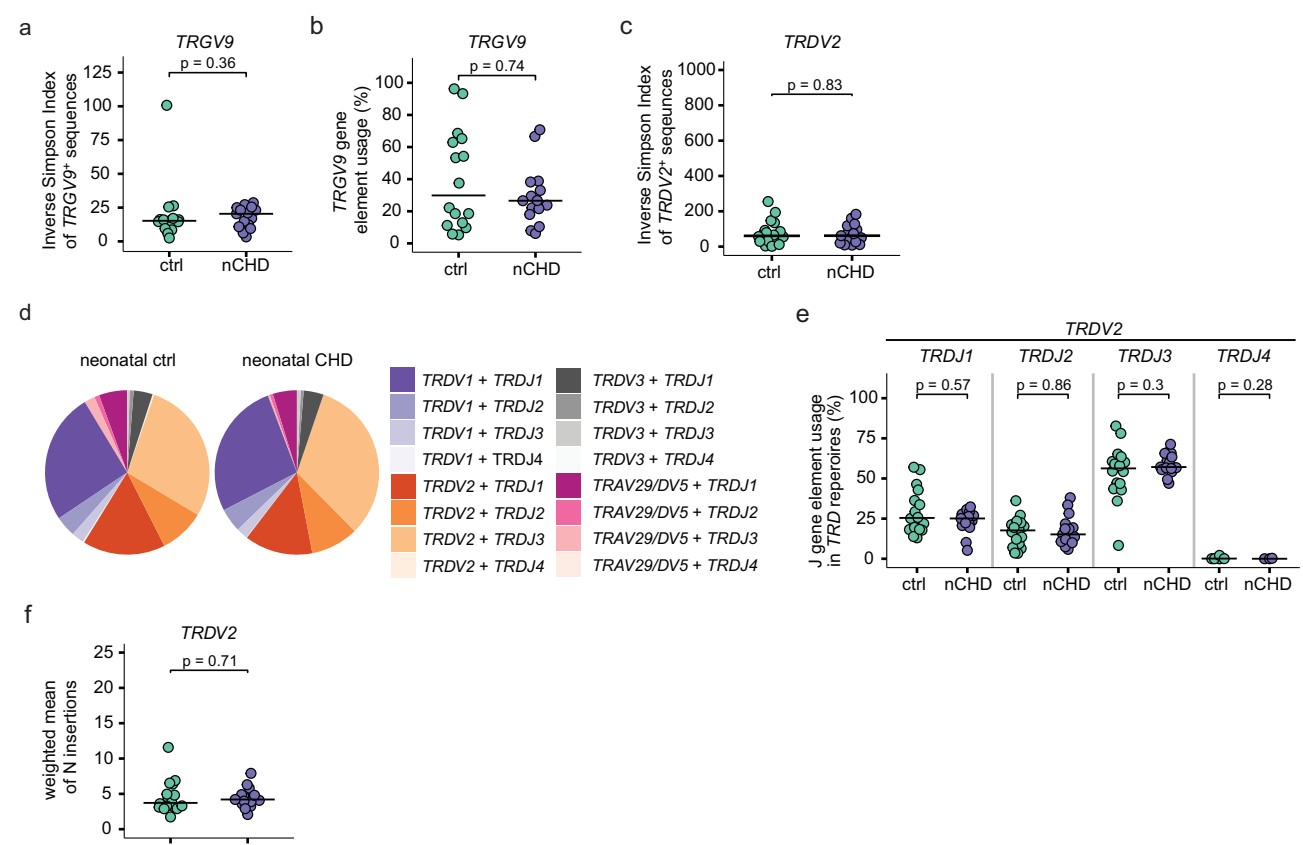

**Fig. 6 | γδTCR repertoires in neonates with CHD but prior cardiac surgery are highly similar to age-matched neonatal controls.** *TRG* and *TRD* repertoire analyses from neonates with CHD prior cardiac surgery (nCHD, n = 15) and neonatal controls (neonatal ctrl, n = 16). **a** Inverse Simpson indices of *TRGV9*+ *TRG* sequences. **b** Proportion of *TRGV9*+ clones within all *TRG* sequences. **c** Inverse Simpson indices of *TRDV2*+ TRD clones. **d** Pie charts, color-coded by *TRDV-TRDJ* pairing within *TRD* repertoires at group-level. **e** J-gene usage in *TRDV2*+ sequences in percent. **f** N-insertion counts in *TRDV2*+ sequences, calculated as weighted mean, weighted by the total number of clones per sample. Statistical analyses were performed using the two-sided Wilcoxon−Mann−Whitney *U* test; not significant (ns). No adjustments for multiple comparisons were made. Bars indicate median values; each dot is representative of one donor.

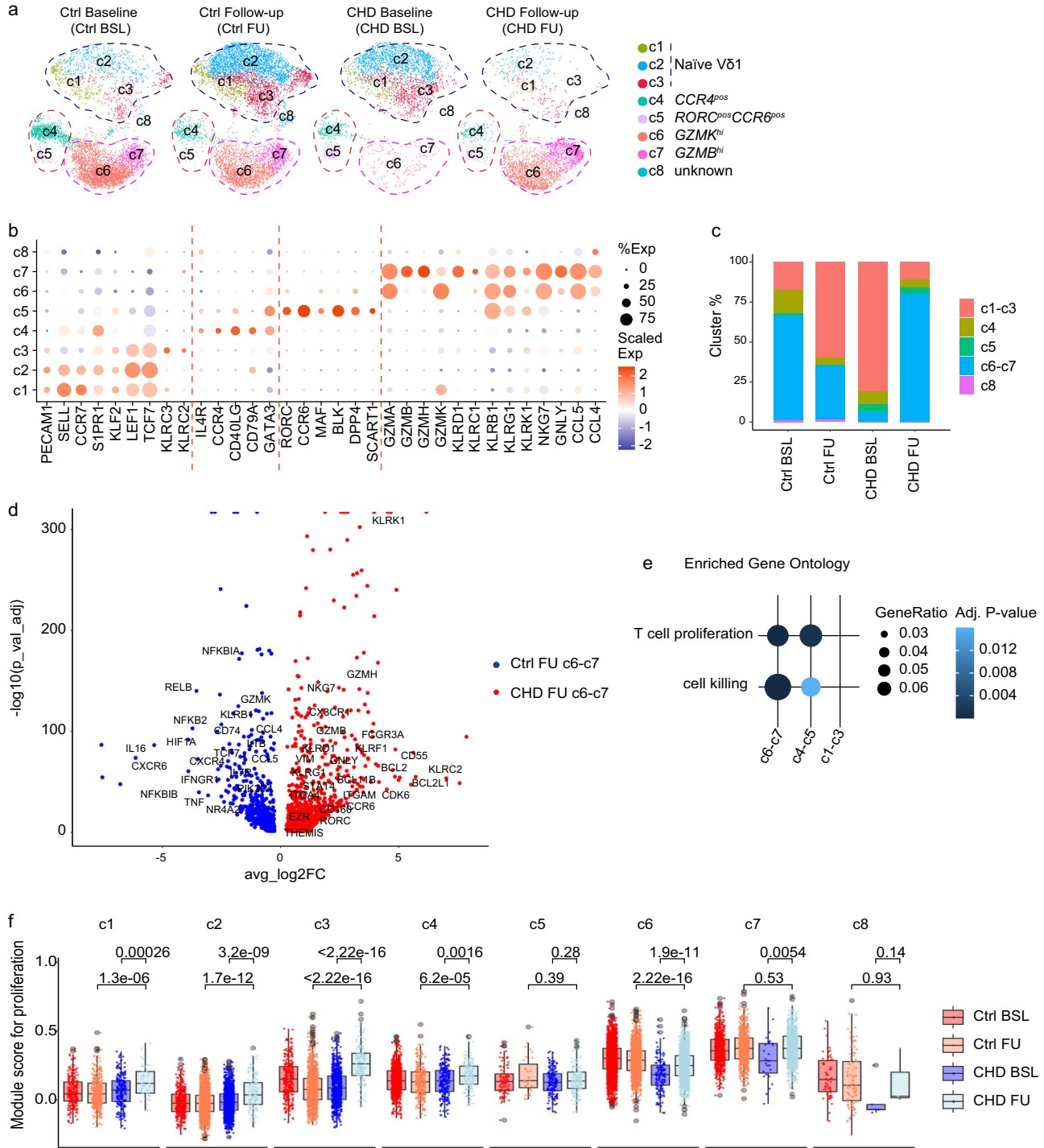

module scores for proliferation were increased in the Vδ1+ clusters c1–c3 and the majority of Vδ2+ clusters at the FU time point (post-surgery) (Fig. 7f). Notably, the analysis indicated that the Vδ2+ clusters c6–7 had the highest proliferative capacity (Fig. 7f). Together, the scRNA-seq data provide a map of the postnatal maturation of γδ T cells in relation to the thymic output in infancy. It shows that GZMB^hi and GZMB^low Vγ9Vδ2 T cells (clusters c6–7), but also Vδ1 T cells, may undergo postnatal proliferation to compensate for reduced postnatal thymic output in infants with CHD who required congenital heart surgery as neonates.

## Discussion

The thymus is the central organ for T cell development and reaches its maximum postnatal output during adolescence[20]. In addition to the age-related physiological involution of the thymus, there remains an ongoing contribution of the thymus to the maintenance of the naïve T cell pool long after puberty[20,49,50]. However, adverse events that disrupt thymic function are associated with immune senescence and impaired immunity, although potential regeneration of residual thymic tissue must also be considered[51]. Many children with CHD require heart surgery as infants. Due to the small size of the body and organs, good visibility and safe access are essential, and the thymus behind the sternum is partially or completely removed (thymectomy) with subsequent changes in thymic architecture and/or size[6]. Perturbations in the αβ T cell phenotypes, expressed TCRs, and disease susceptibility have been reported in thymectomized pediatric and adult patients with CHD[8–10,52]. Studies on γδ T cells are rare in this context[16,35]. Here,

**Fig. 7 | Longitudinal single-cell transcriptomic analysis reveals a proliferative advantage of γδ T cells in infants with CHD at 6 months post-thymectomy.**
**a** UMAP visualization of the γδ T cell scRNA-seq dataset from two longitudinally followed CHD infants before (Baseline, BSL, at age day 8 and 9) and 6 months post-surgery (Follow-up, FU, at age day 188 and 189), colored by the identified clusters c1–c8. Control samples represent transcriptional profiles of age-matched preterm neonates[28]. **b** Dot plot of the average gene expression (columns) per cluster (rows). Dots are colored by average logFC (Scaled Exp) and sized by the percentage of cells per cluster that expressed this gene (% Exp). **c** The bar plot reveals fractions of absolute cell numbers of indicated subpopulations from each group. **d** The volcano plot demonstrates DEGs among the Ctrl FU c6–c7 and CHD FU c6–c7. Upregulated DEGs are identified with log2FC > 0.25 and p_val_adj <0.05. **e** Gene ontology enrichment analysis of the upregulated differential expressed genes from the indicated subpopulations. **f** Box plots of the single-cell gene signature module score for proliferation, based on gene list: *CBLB, PTPN6, IRF1, HLA-DPB1, HLA-DPA1, HLA-DRB1, CCL5, TNFRSF1B, SH2D2A, FYN, BTN3A1, IL12RB1, SPN, CD81, MSN, ANXA1,*

*SASH3, PYCARD, PTPN22, RASAL3, SELENOK, TNFSF14, ITCH, HLA-A, PSMB10, PTPRC, HLA-E, TNFRSF14, IL2RA, CD40LG, TNFRSF4, CR1, ANXA1, XCL1, SOS1, IL6ST, CD55, PRNP,* and *PPP3CA*. The cell numbers of each cluster from each group are as follows: c1, nCtrl BSL = 235, nCtrl FU = 384, nCHD BSL = 215, nCHD FU = 55; c2, nCtrl BSL = 399, nCtrl FU = 2296, nCHD BSL = 1432, nCHD FU = 129; c3, nCtrl BSL = 300, nCtrl FU = 1320, nCHD BSL = 754, nCHD FU = 138; c4, nCtrl BSL = 788, nCtrl FU = 282, nCHD BSL = 229, nCHD FU = 138; c5, nCtrl BSL = 89, nCtrl FU = 31, nCHD BSL = 159, nCHD FU = 127; c6, nCtrl BSL = 2825, nCtrl FU = 1589, nCHD BSL = 141, nCHD FU = 1401; c7, nCtrl BSL = 662, nCtrl FU = 671, nCHD BSL = 27, nCHD FU = 923; c8, nCtrl BSL = 58, nCtrl FU = 56, nCHD BSL = 5, nCHD FU = 3. The minima of boxplot: the lowest data point within 1.5 times the first quartile's Interquartile Range (IQR). Maxima: the highest data point within 1.5 times the IQR of the third quartile. Center: median of the data. Statistical significance was determined by unpaired two-sided Wilcoxon–Mann–Whitney *U* test; not significant (ns). Source data are provided as a Source Data file.

we aimed to investigate the developmental origin, phenotypes and functions of γδ T cells and αβ T cells in a well-defined study population of infants and children with CHD, all of whom underwent corrective or palliative heart surgery within the first 6 weeks of life (median age at surgery = 8.5 days).

A reduction in CD31⁺ naïve αβ T cells, which is an established cellular marker of thymic activity[36,49], reflects the changes in size and/or architecture of the thymic tissue during heart surgery that have not yet been recovered in these children. Flow cytometric analyses also showed a reduction in CD31⁺ Vδ1 T cells in children with CHD. However, a correlation of CD31 expression on γδ T cells with measured T cell receptor excision cycles is missing, and it remains to be seen whether CD31 could serve as a potential marker to define newly generated Vδ1 T cells. For αβ T cells, our data confirm a reduced TCR repertoire diversity, a decrease in naïve CD4 and CD8 T cells, and increased expression of PD-1 and CD57 on T cells in these children[12]. Specifically, PD-1 and CD57 have been described as surface markers of senescent T cells in patients with chronic infections or transplant recipients[38,53–55]. Notably, in vitro TCR stimulation results suggest that the overall responsiveness of CD4 and CD8 T cells was not reduced in the 5- to 12-year-old children with CHD.

Importantly, all children with CHD had undergone surgery that affected the thymic size and architecture prior to the increase of T cell output/development in the infant thymus. This is of particular interest when investigating γδ T cells, which are known to be generated in defined ontogenetic T cells waves in the embryonic, perinatal and postnatal period[19]. In this study, two major peripheral blood γδ T cell populations were analyzed, namely Vγ9Vδ2 T cells (also defined as Vδ2 T cells) and Vδ1 T cells. Vγ9Vδ2 T cells develop as standard effector cells in the prenatal thymus, with only a few being generated in the postnatal period. Conversely, Vδ1 T cells seen in the peripheral blood of adults are most likely of postnatal origin[19]. In infants without thymectomy, diversification of γδTCR repertoires occurs in early infancy[28], most likely related to newly generated T cells (mainly Vδ1 T cells), which have been described to have highly diverse TCR repertoires in the postnatal period. For Vδ1 T cells, thymectomized children with CHD had a reduced overall frequency within lymphocytes and a reduced *TRDV1* TCR repertoire diversity, suggesting that early-life thymectomy disrupted the postnatal thymic burst of naive Vδ1 T cells. In line with this, reduced thymic generation of Vδ1 T cells was evident in the decreased expression of mRNA transcripts related to RTE in both thymectomized infants and children with CHD. To replenish the T cell pool early after thymectomy, remaining or developing Vδ1 T cells presumably undergo homeostatic proliferation, characterized by increased expression of proliferation genes. At the same time, the proliferating remaining Vδ1 T cells have a higher likelihood of encountering antigens that may induce the reduction in TRDV1

repertoire diversity reported here and maturation of naïve Vδ1 T cells into granzyme-releasing effector cells, as evidenced by scRNA-seq and flow cytometry. This leads to the hypothesis that early-life thymectomy may accelerate the maturation into effectors and repertoire focusing of Vδ1 T cells in children with CHD. Vδ1 T cells are known to function as anti-viral and anti-cancer cells in healthy and immunocompromised patients[31]. Future studies in young adults post-puberty, adult and elderly patients with CHD are needed to determine the maturation dynamics of Vδ1 T cells in relation to the thymic output and aging, which will have implications for optimizing anti-cancer therapy and clinical care of this vulnerable patient group.

In contrast to all other T cells, the frequencies of Vγ9Vδ2 T cells were not perturbed in children with CHD who underwent surgery early after birth. The enrichment of GZMA⁺CD161ʰⁱCD28ʰⁱ Vγ9Vδ2 T cells in children with CHD suggests a selective expansion and/or homeostatic maintenance of effector Vγ9Vδ2 T cells after surgery. Importantly, Vγ9Vδ2 T cells were enriched for TCR sequences with low numbers of N-insertions and use of the *TRDJ3* gene element. These TCR features have been reported to be specifically used by early fetal thymus-derived Vγ9Vδ2 T cells[23]. As a proof of concept, and to exclude that the CHD per se leads to T*RDJ3*ʰⁱ *TRD* repertoires, the γδ TCR repertoires were examined in a second study cohort of neonates with diagnosed CHD, but with blood lymphocyte sampling prior to cardiac surgery for CHD, and in age-matched term and preterm control neonates. As the *TRG* and *TRD* repertoires were highly similar in all neonates (independent of prematurity and disease), the enrichment of fetal-derived *TRDJ3*ʰⁱ TRD sequences is likely to be related to the reduced postnatal thymic activity in the 5- to 12-year-old children with CHD. It has been established that the postnatal thymus produces some Vγ9Vδ2 T cells that are high for the *TRDJ1* element[24]. There could be either a bias for preferential use of *TRDJ3*⁺ TCRs during V(D)J recombination and/or intra-thymic selection of *TRDJ3*⁺ γδ T cell clones in the fetal period. In addition, it has been established that Vγ9Vδ2 T cells positive for *TRDJ3* gene elements are rarely seen in adults[39,46]. It has been speculated that Vγ9Vδ2 T cells that are *TRDJ3*⁺ have a lower TCR affinity to respond to pAgs produced by the postnatal microbiota and may therefore be outcompeted by *TRDJ1*⁺ Vγ9Vδ2 T cells during postnatal adaptation[56]. In addition, our scRNA-seq data suggest that early-life congenital heart surgery with an impaired postnatal thymic T cell output supports the homeostatic expansion of Vγ9Vδ2 T cells (presumably being *TRDJ3*ʰⁱ) and Vδ1 T cells in early infancy. Despite this, Vγ9Vδ2 T cells and Vδ1 T cells in infants showed mild changes in transcriptional profiles around 6 months after CHD surgery. Of note, scRNA-seq of peripheral blood γδ T cells from preterm infants served as a non-CHD control in this experimental setting. However, we expect that especially in the 6- to 8-month-old infants, γδ T cells

will show similar gene expression profiles between term and pre-term infants, as has been reported for other immune cell parameters[57].

Transcriptional changes were not apparent until later in childhood, and increased expression levels of cytotoxicity genes were found in children with CHD. A subset of γδ T cells with gene expression profiles of fetal-derived γδ T cells (e.g., being *PLZF*hi, *CD28*hi and *GZMA*+) was elevated in children with CHD. Transcriptional and flow cytometric data provide evidence that the innate CD161hiCD28hi Vγ9Vδ2 T cells retained their functional capacity with a bias toward GZMA production, which has been described to be predominantly produced by fetal-derived γδ T cells[27,58], in the children with CHD. Notably, the identified scRNA-seq cluster c6 showed high expression for *CD8A* and *CD8B* (RNA level only), which are associated with perinatally derived γδ T cell effectors[59]. Second, transcripts described as expressed by human γδ T cells in tissues were identified in the abundant cluster c6 in the older children with CHD, but not in any of the enriched clusters in infants with CHD[44]. Thus, alterations in the postnatal thymic activity may have more long-term consequences. In addition, it remains to be seen whether these innate γδ T cells are re-circulated from the tissues of the thymectomized older children. Overall, we conclude that early-life thymectomy disrupts the transcriptional programs of Vγ9Vδ2 T cells in the long term and leads to an increased frequency of GZMA+CD161hiCD28hi Vγ9Vδ2 T cells in children. Future investigations are needed to understand whether the predominantly fetal thymus-derived Vγ9Vδ2 T cells in adults with CHD do not undergo immune senescence or may do so later in life.

### Limitations of the study

This study provides robust and consistent data on the postnatal adaptation and persistence of fetal and postnatal γδ T cells in small, but homogeneous study populations of children and neonates with CHD and without CHD. In particular, the study provides insights into the postnatal maturation of γδ T cell subsets and their functions in relation to the postnatal thymic output. Of note, thymectomy was confirmed by reviewing surgical reports of surgery for CHD in the neonatal period, which makes it difficult to predict the extent of surgical removal of thymic tissue and its postoperative regeneration potential. Therefore, we included the CD31 surface expression to reliably determine the postoperative thymic activity. Using different techniques, we robustly identified an enrichment of GZMA+ CD28+ CD161hi effector Vγ9Vδ2 T cells in children with CHD who had undergone cardiac surgery with thymectomy shortly after birth. Remarkably, Vδ1 T cells were severely depleted after thymectomy, and those that remained had cytotoxic effector functions. Indeed, thymectomy may lead to disease susceptibility later on[8–10,52,60,61]. Thus, the long-term perspective of the regeneration of thymic activity, immune cell phenotypes and the T cell functionality, as well as inter-individual differences between CHD patients and how these depend on the age, remain to be investigated.

## Methods
### Study cohorts
This observational single-centered study was conducted in accordance with the Declaration of Helsinki and was approved by the institutional ethics board at Hannover Medical School (10198_BO_S_2022). Written informed consent was obtained for each child from their legal guardians, and, depending on age, the child as well.

Our study focused on investigating immunological changes in children who had undergone cardiac surgery of CHD with concomitant thymectomy and enrolled 5- to 12-year-old children after surgery (Table 1) and neonates before surgery (Table 3). The children and neonates were diagnosed with a wide spectrum of critical and/or complex CHDs, such as simplex d-TGA (dextro-transposition of great

arteries), complex TGA, TAPVR (total anomalous pulmonary vein return), and Shone's variant (Supplementary Tables S1, S2), in some cases requiring multiple surgical and/or interventional procedures.

For the post-surgical study population, children with CHD (aged 5–12 years) who visited the pediatric cardiology outpatient clinic for a routine follow-up appointment, were eligible for the study. These study participants underwent partial or complete thymectomy within the first 6 weeks of life performed during palliative or corrective heart surgery (Supplementary Table S1). Age-matched, generally healthy children who had a blood sample taken before planned minor surgery at the Department of Pediatric Surgery, Hannover Medical School, Hannover, Germany, were included in the study as non-CHD controls.

The second study population consisted of neonates with CHD ($n = 15$). Here, biosamples were taken before planned congenital heart surgery. Healthy non-CHD 2- to 3-day-old term neonates were sampled during routine blood collection as part of the newborn screening at the Department of Obstetrics, Gynecology and Reproductive Medicine, Hannover Medical School (MHH), Hannover, Germany. All included patients had no diagnosed genetic disorders, diagnosed or suspected primary and secondary immunodeficiencies, or infections at sampling with the following exception: one 6-year-old child with CHD was diagnosed with heterotaxy syndrome, with right isomerism. Clinical data were obtained through a standardized questionnaire completed by the parents at the study visit, and the patients' health records.

Additionally, we used sequencing data on the *TRD* and *TRG* repertoire of preterm neonates (born at a gestational age of 26 to 32 weeks) previously reported in ref. 48, and transcriptional profiles of peripheral blood γδ T cells from two longitudinally followed uninfected preterm neonates without CHD (Ctrl at BSL and FU, GSE245131[28]).

### Isolation of peripheral mononuclear blood cells
Peripheral mononuclear blood cells (PBMCs) were freshly isolated from EDTA blood following density medium centrifugation. Isolated cells were gently frozen in a freezing medium containing 90% heat-inactivated fetal calf serum and 10% DMSO, and stored in aliquots at −80 °C until further processing.

### Multicolor flow cytometry and FACS
PBMCs were thawed in a 37 °C water bath, prepared by washing in 0.3% FCS (Sigma®), 3 mM EDTA (Carl Roth GmbH + Co. KG), PBS buffer, and apportioned in $1 \times 10^6$ cells per participant for the subsequent spectral flow cytometric analysis, and in $>1 \times 10^6$ cells for sorting of γδ, CD4 and CD8 T cells. After thawing, samples were stained for 20 min at room temperature with an antibody mix containing 24 monoclonal antibodies and a viability dye for cell phenotyping assessment of functional parameters, including negative checkpoint regulators (NCRs) and co-stimulatory molecules (Supplementary Table S3).

Acquisition, unmixing and compensating were performed on the spectral flow cytometer Cytek Aurora using Spectroflo software (Cytek Biosciences). Subsequent gating of cell populations of interest was performed using FCS express software (version 7.14.0020). For isolation of described populations on an FACS Aria™ Fusion (BD), thawed PBMCs were stained with antibodies directed against the following surface antibodies: CD3 (PE-Cy7, SK7, BD, 1:50), TCRγδ (PE, REA591, Miltenyi, 1:50), TCRαβ (FITC, WT31, BD, 1:50), CD4 (PerCP, M-T466, Miltenyi, 1:50), CD8 (APC-Cy7, SK1, BioLegend, 1:20), CD25 (BV605, BC96, BioLegend, 1:50), CD127 (APC, MB15-18C9, Miltenyi, 1:50), iNKT (BV786, B11, BioLegend, 1:50); dead cells were detected via staining with Zombie DAPI. Sorted γδ T cells (Zombie DAPI−, CD3+, TCRγδ+) were resuspended in RLT RNeasy Lysis Buffer (Qiagen). Sorted CD4+ (Zombie DAPI−, CD3+, CD4+, non-CD25hi/CD127lo) and sorted CD8+

T cells (Zombie DAPI⁻, CD3⁺, CD8⁺) were spun down and supernatant was discarded. All sorted populations were frozen and stored at −80 °C.

For intracellular staining of GZMA/GZMB, thawed PBMCs were first stained with a cocktail of surface antibodies on ice for 20 min, followed by fixation and permeabilization with Foxp3 staining kits (eBioscience), according to the corresponding protocol. The surface antibodies and intracellular antibodies were used as follows: CD8A (BUV805, SK1, 1:100), Vd1 (VioGreen, REA173, 1:50), CD4 (BV750, SK3, 1:200), CD3 (AF532, SK7, 1:25), CD8B (FITC, S21011A, 1:50); Vδ2 (PerCP-Vio700, REA771, 1:200), TCRγδ (PE, REA591, 1:200), CD28 (PE-Cy5, CD28.2, 1:100), GZMA (APC, CB9, 1:50), GZMB (AF647, GB11, 1:50), CD19 (APC-Fire810, HIB19, 1:100) (Supplementary Table S6).

## TCR-seq

mRNA extraction was performed from thawed FACS-sorted alive/CD3⁺ γδ T cells, CD4 T cells and CD8 T cells according to instructions of the RNeasy mini kit (Qiagen). For TRG and TRD repertoire analyses, the isolated mRNA was reverse-transcribed into cDNA using Superscript III enzyme (Invitrogen) and oligo(dT)primers. Amplification of CDR3 regions for TRG clones and TRD clones was performed using a multiplex polymerase chain reaction (PCR). PRIMER sequences target major Vγ-genes or Vδ-genes and the respective constant gene regions, as described previously with the following gene-specific primer sequences: h*TRD*V1: TCAAGAAAGCAGCGAAATCC; h*TRD*V2: ATTGCAAA GAACCTGGCTGT; h*TRD*V3: CGGTTTTCTGTGAAACACATTC; h*TRD*V5/ 29: ACAAAAGTGCCAAGCACCTC; h*TRD*C1: GACAAAAACGGATGG TTTGG; h*TRG*V(2,3,4,5,8): ACCTACACCAGGAGGGGAAG; h*TRG*V9: TCGAGAGAGACCTGGTGAAGT; h*TRG*C(1,2): GGGGAAACATCTGCAT CAAG[39]. For TRB repertoire analysis, the isolated mRNA was reverse-transcribed using SMARTer RACE 5′-3′ PCR Kit (Clontech) with a customized protocol[62]. Next, the CDR3 region of TRB clones was amplified via RACE PCR using a gene-specific primer for the constant gene region (GCACACCAGTGTGGCCTTTTGGG). For all conditions and to control PCR contamination, negative controls (H2O) were run together with each PCR, and also subjected to sequencing. All PCR amplicons were subsequently prepared for paired-end Illumina sequencing (v2 500 cycles)[39], using the Illumina MiSeq platform according to the Illumina guidelines. For the preterm neonates, the available *TRG* and *TRD* repertoires were taken from our previous study (Bioproject PRJNA592548)[48].

## TCR data analysis

Sequencing reads from obtained FASTQ files were aligned and annotated with MiXCR 4.3.2 according to the international immunogenetics information system[63]. Annotated files were analyzed with V(D)J tools and the Immunarch 1.0.0 package in R[64].

## High-dimensional flow cytometry data analysis

For unsupervised cluster analysis of γδ T cells, flow cytometry sample files were first pre-gated on γδ T cells (CD3+TCRγδ+) in FCS Express and exported as.fcs files. Those were then loaded into R and transformed using the Logicle transformation[65]. Unbiased automatic detection and removal of suspected anomalies in the flow cytometry data were performed with the Peak-based selection of high-quality cytometry data (PeacoQC, 1.14.0) algorithm[66]. The cleaned cytometry data were then subjected to the batch-correction algorithm cyCombine[67] to adjust for possible batch effects due to the different time points of data acquisition. Next, we employed the Flow Self-Organizing Maps (FlowSOM, 2.12.0) algorithm[68] to perform unsupervised clustering of cells based on their global marker expressions of Vδ1, Vγ9, Vδ2, CD8, CD4, CD45RA, CCR7, CD28, CD27, CD25, CD127, CD25, CD127, CD161, NKG2A, CD31, CD57, PD-1, CD16, and CD56. The number of meta-clusters was set to 8. Of those, one small cluster did not express any γδ T cell-specific lineage markers but was instead

highly CD4⁺ and, therefore, deemed a contamination, removed from this analysis. The remaining cells were re-clustered with FlowSOM to obtain 8 clusters of TCRγδ positive cells. Dimensionality reduction was performed using the Uniform Manifold Approximation and Projection (UMAP) approach. Cell clusters were annotated based on their respective marker expression characteristics. Finally, the relative abundance of each cluster was calculated as a percentage of all CD3⁺TCRγδ⁺ cells and used for statistical analyses.

## scRNAseq library and bioinformatics flow

Peripheral blood mononuclear cells (PBMCs) were obtained from four 5- to 11-year-old Congenital Heart Disease (CHD) patients who underwent cardiac surgery with thymectomy shortly after birth, four age-matched non-CHD controls, and two CHD infants both before (Baseline, BSL, at age day 8 and 9) and 6 months after thymectomy (Follow-up, FU, at age day 188 and 189) (Supplementary Table S2). The PBMCs were thawed, and fluorescence-activated cell sorting (FACS) was performed to isolate live CD3+ cells. These sorted T cells were then used to generate single-cell RNA sequencing (scRNAseq) libraries with the Chromium Next GEM Single Cell V(D)J Reagent Kits v1.1 (for infants) and v2 (for 5- to 11-year-old children), following the manufacturer's protocol (10x Genomics). The scRNAseq libraries were sequenced on the NovaSeq X platform, targeting 25,000 read pairs per cell. Reads were aligned to the reference genome GRCh38, and the gene expression matrix was generated using Cell Ranger 7.0 (10x Genomics). Subsequent analysis was performed on γδ T cells using Seurat v5.0.1[69] in R 4.3.2. For comparative analysis, an age-matched non-CHD public γδ T cell dataset (GSE245131[28]) was used to compare CHD patients before and 6 months after thymectomy. Batch effects were corrected using the R package Harmony[70]. Low-quality cells were identified and excluded based on mitochondrial gene content over 10%, or detected genes numbering fewer than 200 or more than 4000. Datasets were merged, normalized, and scaled, followed by PCA and dimension reduction using RunPCA and RunUMAP functions, respectively. The data were then clustered and characterized based on gene expression profiles of lineage markers. Differentially expressed gene analysis was performed with the Seurat function FindMarkers/FindAllmarkers. Gene expressed in at least 10% of the clusters and having p_val_adj <0.05 (BH correction) were considered significant. The gene list was then subjected to the enriched gene ontology test with enrichGO function in R package clusterProfiler[71].

## In vitro stimulation and functional assays

For stimulation experiments, PBMCs were thawed, washed and resuspended in advanced RPMI medium (RPMI 1640 (Gibco)), supplemented with 10% heat-inactivated FCS (Sigma), 1% Penicillin-Streptomycin (Gibco), 1% GlutaMAX (Gibco) and $3.5 \times 10^{-6}$ % β-Mercaptoethanol (Sigma). Cells were incubated for 4 h under humidity, enabling gas exchange. After resting, cells were labeled with cell tracer blue (Invitrogen) to examine proliferation as control. Cell numbers were determined, and cells were cultured at $2 \times 10^5$ cells per well in a 96-well U-bottom plate using IL-2 100 IU/mL (Sigma) supplemented advanced RPMI medium and with the respective stimulant. For HMBPP stimulation, 1 μM HMBPP was added, and cells were cultured at 37 °C for 7 days. For anti-CD3 stimulation, 96-well U-bottom plates were coated with anti-CD3 (UCHT1, BioLegend) and anti-CD28 (CD28.2, BioLegend) and incubated for 4 h. Subsequently, $2 \times 10^5$ cells were added for 48 h stimulation. For each stimulation (HMBPP 7d, and anti-CD3/anti-CD28 48 h), a negative control with IL-2 medium only was also conducted. Staining for surface markers and subsequent intracellular staining were performed with the markers listed in Supplementary Tables S4 and S5. Expressed surface molecules, intracellular cytokine and frequencies of Vγ9Vδ2 T cells and CD4/CD8 T cells were determined using flow cytometry. Acquisition, unmixing and

compensating were performed on the spectral flow cytometer Cytek Aurora, as described for the phenotyping experiment.

## Statistical analysis

Patient demographics and clinical data were summarized using descriptive statistics. Group comparisons of cell frequencies and counts between CHD and control participants were performed using the Mann–Whitney $U$ test. Correlations were evaluated using Spearman's method. In all statistical tests, a $p$-value of <0.05 was considered statistically significant. No adjustment for multiple testing was applied due to the exploratory nature of this study. All analyses were conducted with the statistical software R (version 4.0.3).

## Reporting summary

Further information on research design is available in the Nature Portfolio Reporting Summary linked to this article.

## Data availability

Raw FASTQ files of the TCR-seq data have been deposited at the NCBI SRA under the BioProject: (https://www.ncbi.nlm.nih.gov/bioproject/?term=PRJNA1040021). Raw and processed scRNAseq data are accessible under NCBIs Gene Expression Omnibus (https://www.ncbi.nlm.nih.gov/geo/query/acc.cgi?acc=GSE274083). Source data are provided with this paper.

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

## Acknowledgements

We thank Dr. Matthias Ballmaier and the sorting facility at Hannover Medical School for their kind support in performing FACS. Hannover Medical Research Schools supported A.C., X.L.L., T.Y., L.R. and V.A. The work was supported by the German Research Foundation Deutsche Forschungsgemeinschaft (DFG) under Germany's Excellence Strategy, EXC 2155 RESIST, Project ID 390874280 to R.F., and S.R.; and the DFG-funded research group FOR2799 Project ID RA3077/1-2 to S.R. and *kinderherzen* Fördergemeinschaft Deutsche Kinderherzzentren e.V. to M.B., and S.R.

## Author contributions

Study concept and design: A.C., T.Y., S.R., M.B.; recruitment and sample collection: A.C., E.G., S.K., C.K., A. Horke, A. Hofmann, C.v.K., M.B.; experiments: A.C., T.Y., A.J., Y.A., V.A., X.L.-L.; data analysis: A.C., L.R., T.Y.; data interpretation: A.C., T. Y., P.B., R.F., M.B., S.R.; data visualization: A.C., T. Y., L.R.; statistics: A.C., L.R.; supervision: M.B., S.R.; writing of the initial manuscript: A.C., T.Y., S.R. The manuscript was read and approved by all authors.

## Funding

## Competing interests

The authors declare no competing interests.
