## [Peer Review File · Nature Communications]

Early-life thymectomy leads to an increase of granzyme-producing $\gamma\delta$ T cells in children with congenital heart diseaseREVIEWER COMMENTS

Reviewer #1 (Remarks to the Author):

The manuscript by Ravens and colleagues provides new insights into the pool of gamma delta T cells in children who have had their thymus partially or full removed early in life due to congenital heart disease (CHD). The data is interesting and is a nice addition to recent publications that describe the origin of V δ 2+ $\gamma\delta$ T cells. While the frequency of total V δ 2+ $\gamma\delta$ T cells is unchanged in individuals that underwent thymectomy versus controls, there was a subtle increase in CD161hi NKG2A+ CD28hi V γ 9V δ 2 T cells in thymectomized children. A major finding of this paper is an increase of fetal-derived TRDJ3+ V δ 2+ $\gamma\delta$ T cells in children with CHD that had undergone thymectomy. Collectively, these data reveal that fetal-derived V δ 2+ $\gamma\delta$ T cells can expand in children to a similar extent to V δ 2+ $\gamma\delta$ T cells that develop after birth. The data are also in line with recent reports that suggest that TCR repertoire of V γ 9V δ 2 T cells is altered after birth, highlighting an important role for the post-natal thymus in the selection and development of V γ 9V δ 2 T cells.

Comments:

- The authors gate on gamma delta T cells as a proportion of total leukocytes (including monocytes). Do the results of their study differ by simply gating on total lymphocytes.
- In Figure 1, why was CD31 expression not studied for CD8+ T cells?
- "minor changes in the number of nucleotide insertions within the variable V(D)J regions in the CHD children as compared to non-CHD controls (Suppl. Fig. 1C)" This was only observed for CD4+ T cells.
- "The expressed $\gamma\delta$ TCR V gene can classify human $\gamma\delta$ T cells into V γ 9V δ 2 T cells, which predominantly develop in the fetal thymus, and V δ 1 T cells, which are the dominant $\gamma\delta$ T cell population in the postnatal thymus.24,40,41." This statement is misleading, with recent studies highlighting that adult V γ 9V δ 2 T cells are likely derived from the post-natal thymus.
- The flow cytometry data presented in the manuscript needs to be carefully re-analysed. A representative flow cytometry gating strategy should be provided as supplementary data, with examples of all markers shown.
- The authors suggest there are no differences between the clusters in control versus CHD samples, although cluster C6 (Vd1+ cells) appears to be exclusively observed in the control sample (Figure 3B).
- The data in Figure 3C (and supplementary 3C) suggests some level of CD4 expression by V γ 9V δ 2 T cells, which does not quite fit with what is known about these cells. Similarly, cluster 1 in Figure 3D suggests that many V γ 9V δ 2 T cells are positive for CD8 expression. The addition of conventional flow cytometric analysis would be complementary to the current data set.
- Analysis of cytokine expression by V γ 9V δ 2 T cells can be achieved by short term stimulation with PMA and Ionomycin. Long term stimulation with HMBPP or anti-CD3/28 may obscure subtle differences between CHD and control groups. Granzyme B can also be analysed direct ex-vivo without the need for in vitro stimulation.
- Greater discussion should focus on the origins of V γ 9V δ 2 T cells. The data provided by the authors highlight the importance of the thymus in shaping the pool of V γ 9V δ 2 T cells after birth, yet this is poorly discussed in the context of recent papers in the field. The presence of TRDJ1+ V γ 9V δ 2 T cells in post-natal thymus and not fetal thymus (Papadopoulou J Immunol 2019) suggests that selection in the thymus is somehow altered after birth, rather than preferential expansion of TRDJ1+ V γ 9V δ 2 T cells in response to microbial exposure.

Reviewer #2 (Remarks to the Author):

Recent work has explored the immunological consequences of thymectomies in infancy on adult immune function, and identified reduced numbers and repertoire diversity of $\alpha\beta$ T cells. However, the consequences of partial/full thymectomy on non-conventional T cell lineages, specifically those that differentiate during exclusive developmental windows, remains unknown.

Here, authors investigate the effects of partial/full thymectomy in CHD children on the development and persistence of $\gamma\delta$ T cell subsets. Authors focus on the two major thymically derived $\gamma\delta$ T cell subsets present in human blood; prenatally derived V γ 9V δ 2 T cells and postnatally derived V δ 1+ T cells. Comparison of isolated peripheral blood lymphocytes (PBMcs) between control groups and thymectomized infants with CHD, 5–12-year post-surgery, revealed reduced frequency and repertoire diversity of V δ 1 T cells and elevated frequencies of fetal-derived V γ 9V δ 2 T cells. Overall, these findings suggest that infant thymectomy results in a long-term reduction in postnatally derived V δ 1 T cells in peripheral blood.

Given the immunological importance of $\gamma\delta$ T cells in infection and anti-tumor responses, the findings of this manuscript are undoubtedly of high interest to the field, offering developmental insight into these non-conventional thymically derived T cell subsets. However, given the limitations of this study, including those mentioned by the authors at the end of the manuscript, and the overall volume, depth and complexity of the results presented, I do not think the manuscript as it currently stands contains enough novel data for publication in a journal of this impact. If authors could include more insightful functional analysis, explore the longer-term effects of this observed altered $\gamma\delta$ T cell heterogeneity in older cohorts, or delve deeper into potential alterations to additional $\gamma\delta$ T cell subsets, at both a protein and transcriptional level, this would strengthen the manuscript's conclusions greatly and would widen its impact.

Focused points:

The manuscript uses flow cytometry and TCR sequencing approaches to examine the differences in $\gamma\delta$ T cells in PBMcs from CHD patients and controls. In addition, authors perform in-vitro stimulation assays to explore functional alterations between V γ 9V δ 2 T cells from the two groups. Overall, the experiments displayed seem well executed, the results are clearly displayed and presented to a high standard. However, in many figures the data displayed could be explored in greater depth, representative flow plots are absent, vital control group comparisons are missing and only specific $\gamma\delta$ T cell subsets are examined, all at a frequency level.

Figure 1

Figure 1 A to H confirm known changes in $\alpha\beta$ T cells post thymectomy. The biological significance of many of the populations explored is absent from the text, including TEMRA populations and increased PD1 on T cells. No representative flow plots are displayed. Figure 1G onwards explores $\gamma\delta$ T cell subsets by flow cytometry. Representative flow plots should be displayed and Figure 1L could be presented as a ratio to give more clarity on the proportional change of V γ 9V δ 2 vs all other $\gamma\delta$ T cell subsets.

Figure 2

A and B: The data suggested a higher frequency of TRGV9 but a reduced Simpson's clonality score, suggesting reduced diversity. Could this be a result of clonal expansion of the increased V γ 9V δ 2 T cell population? Could authors explore proliferation rates, either by flow with Ki67, and/or at a transcriptional level, comparing expression of cell cycle genes, to speculate why this change might be occurring. I also think that means conclusions like "Taken together, $\gamma\delta$ TCR repertoire analyses depict a reduction of clonal diversity, which is restricted to the V δ 1 T cell subset" are inaccurate, as differences in the diversity of the TRGV9 are clearly displayed.

C and E: The data included in the treemaps from two representative donors appear to display differences in many V-gene elements and J-gene elements between the ctrl and CHD groups. However, the intricacies of these changes are hard to identify as the data is currently displayed. Additional graphs (for example pie graphs) could explore this in more depth and the inclusion of all samples would strengthen findings across all individuals.

G: No control graph is displayed. Do authors see the same correlation in control groups with the natural variation in %RTE between individual's samples?

Figure 3

Authors use a UMAP dimension reduction approach to explore in more depth flow cytometric differences displayed in figure one. Clusters have been labelled V δ 1 or V δ 9, based on the expression of markers on a scaled heat map (figure 3C). Does the heat map display the same clear trends if unscaled? Including feature plots for the most important proteins, such as V δ 1, V δ 9 and V δ 2 would add confidence to the clusters analysed. Additionally, a bar graph for the frequency of Ctrl and CHD cells per cluster would help confirm changes between clusters. This would add to the analysis in figure 3E, where the frequency of all $\gamma\delta$ T cell is broken down per cluster. The histograms of each marker in 3D do not allow exploration of co-expression of the surface markers. Displaying flow plots of the key markers against each other, such as CD31 vs CD8, would enable dual expression to be explored further. Figure 3E identifies phenotypical differences of V δ 1 T cells

between the two groups, with lower CD31^{hi} (with and without CD8) frequencies in the CHD group. Authors propose CD31 as a potential marker for recent thymic V δ 1 T cell emigrants, which is reduced in the CHD group. However, no data is displayed to support CD31 as a RTE marker on human $\gamma\delta$ T cells.

Figure 4:

No in-vitro functional assessment of the V δ 1 T cells is performed, although this population displays the biggest difference between control and CHD groups. Representative flow plots from the control groups are also missing for the V γ 9V δ 2 T cells (figure 4A).

Reviewer #3 (Remarks to the Author):

Cramer et al investigated the impact of neonatal thymectomy in patients with congenital heart disease (CHD) on the persistence of $\gamma\delta$ T cells. The authors employed both flow cytometry and TCR repertoire analysis to determine how $\gamma\delta$ T cell subsets were impacted. They report that while postnatally produced V δ 1 $\gamma\delta$ T cells were diminished in thymectomized patients, fetal-derived V γ 9V δ 2, and specifically the CD161^{hi}NKG2A⁺ subset showed elevated abundance. They also suggest that the study provides insight into how loss of the thymus is associated with changes in persistence fetally-derived $\gamma\delta$ T cells. The experiments are well designed and executed; however, there are a couple points that diminish enthusiasm.

First, the data do not support the assertion that V γ 9V δ 2 cells are increased in abundance. Panel 1K clearly shows that when normalized to leukocytes, the V γ 9V δ 2 representation is unchanged, despite there increased representation among $\gamma\delta$ cells.

Second, while the analysis is very careful, the results are somewhat expected in that V γ 9V δ 2 cells that are primarily fetally-derived persist after thymectomy, while postnatally produced V δ 1 cells are diminished as would be predicted.

Point by point reply

Reviewer #1 (Remarks to the Author):

The manuscript by Ravens and colleagues provides new insights into the pool of gamma delta T cells in children who have had their thymus partially or full removed early in life due to congenital heart disease (CHD). The data is interesting and is a nice addition to recent publications that describe the origin of V δ 2+ $\gamma\delta$ T cells. While the frequency of total V δ 2+ $\gamma\delta$ T cells is unchanged in individuals that underwent thymectomy versus controls, there was a subtle increase in CD161^{hi} NKG2A⁺ CD28^{hi} V γ 9V δ 2 T cells in thymectomized children. A major finding of this paper is an increase of fetal-derived TRDJ3+ V δ 2+ $\gamma\delta$ T cells in children with CHD that had undergone thymectomy. Collectively, these data reveal that fetal-derived V δ 2+ $\gamma\delta$ T cells can expand in children to a similar extent to V δ 2+ $\gamma\delta$ T cells that develop after birth. The data are also in line with recent reports that suggest that TCR repertoire of V γ 9V δ 2 T cells is altered after birth, highlighting an important role for the post-natal thymus in the selection and development of V γ 9V δ 2 T cells.

Answer: We thank the reviewer for the overall positive, but critical feedback. We are grateful that the question on the functionality of V γ 9V δ 2 and V δ 1 T cells post-thymectomy has been raised. As outlined in the revised manuscript and our reply below, newly generated scRNA-seq data of $\gamma\delta$ T cells prior and post thymectomy, and flow cytometric assessment of give now insight in the functional properties of $\gamma\delta$ T cells, namely increased GZMA production of V γ 9V δ 2 and GZMB and GZMA release of V δ 1 T cells in children with CHD, which received CHD surgery as neonates.

Comments:

The authors gate on gamma delta T cells as a proportion of total leukocytes (including monocytes). Do the results of their study differ by simply gating on total lymphocytes.

Answer: We appreciate the reviewer's critical analysis of our data. In the revised version, we analyzed the frequency of $\gamma\delta$ T cells within lymphocytes (**new Fig. 1g**). The results were similar to those obtained when analyzing $\gamma\delta$ T cells within leukocytes, as depicted in the **Reviewer Figure 1.1**. To be consistent in the revised manuscript results, we have additionally analyzed CD3, CD4 and CD8 T cell frequencies among lymphocytes (**revised Fig. 1**).

Reviewer Figure 1.1. Dot plots showing the percentage of $\gamma\delta$ T cells within pan-leukocytes (A), within total lymphocytes (B) and within (C) CD3⁺ cells per patient. Statistical analyses were performed using the Mann-Whitney U test (ns $p > 0.05$, * $p < 0.05$, ** $p < 0.01$, *** $p < 0.001$,

**** $p < 0.0001$). Bars indicate median values, with each dot representing a single donor.

In Figure 1, why was CD31 expression not studied for CD8+ T cells?

Answer: CD31 has only been described as a marker for recent thymic emigration on naïve CD4 T cells (Ao et al., 2022; Drozdov et al., 2022; van den Broek et al., 2018). It is unclear whether CD31 expression has a similar significance on naïve CD8 T cells. Thus, we prefer to only include CD31 on naïve CD4 T cells as a value for RTE in the manuscript. Nonetheless, a similar trend is observed on CD8 and on CD4 T cells, as shown in **Reviewer Figure 1.2**. However, it should be emphasized that previous studies have questioned the use of CD31 to identify CD8 T cells that have recently emigrated from the thymus, as it is not a unique marker.

Reviewer Figure 1.2. Percentage of CD31⁺ cells within naïve (CD45RA⁺CCR7⁺) CD8 T cells and naïve (CD45RA⁺CCR7⁺) CD4 T cells, the latter labelled as Recent Thymic Emigrant (RTE). Statistical analyses were performed using the Mann-Whitney U test (ns $p > 0.05$, * $p < 0.05$, ** $p < 0.01$, *** $p < 0.001$, **** $p < 0.0001$). Bars indicate median values, with each dot representing a single donor.

“minor changes in the number of nucleotide insertions within the variable V(D)J regions in the CHD children as compared to non-CHD controls (Suppl. Fig. 1C)”. This was only observed for CD4+ T cells.

Answer: Thank you for pointing this out. The text has been amended accordingly: *“The number of nucleotide insertions within the variable V(D)J regions was marginally lower in the CHD children compared to the non-CHD controls, exclusively for the TRB repertoire of CD4 T cells, but not CD8 T cells (Supplementary Figure 1h)”*. Overall TRB repertoire analysis depicted similar changes as described by previous studies (Gudmundsdottir et al., 2016; Kooshesh et al., 2023).

“The expressed $\gamma\delta$ TCR V gene can classify human $\gamma\delta$ T cells into V γ 9V δ 2 T cells, which predominantly develop in the fetal thymus, and V δ 1 T cells, which are the dominant $\gamma\delta$ T cell population in the postnatal thymus.^{24,40,41”} This statement is misleading, with recent studies highlighting that adult V γ 9V δ 2 T cells are likely derived from the post-natal thymus.

Answer: We thank the reviewer for pointing this out. Indeed, V γ 9V δ 2 T cells can be generated in the postnatal thymus, albeit they are the minor $\gamma\delta$ T cell subset to be generated in this time window. We have revised the text accordingly: *“There was a reduction of $\gamma\delta$ T cells among lymphocytes in the children with CHD that underwent thymectomy as neonates (Fig. 1g). The expressed $\gamma\delta$ TCR can classify human $\gamma\delta$ T cells into V γ 9V δ 2 T cells, which predominantly develop in the fetal thymus, and V δ 1 T cells. The V γ 9V δ 2 T cells are one of the first major T cell subset in the early fetal thymus,⁴⁴ but can still be*

generated by the postnatal thymus.²⁶ V δ 1 T cells are the dominant $\gamma\delta$ T cell population in the postnatal thymus.^{24,45,46}

The flow cytometry data presented in the manuscript needs be carefully re-analysed. A representative flow cytometry gating strategy should be provided as supplementary data, with examples of all markers shown.

Answer: Many thanks for the suggestion. In the **reviewer Figure 1.3** we now provide gating strategies for the identification of immune cell subsets, and have stated in the revised Figure legends the surface markers used for the immune cell subset. In the revised version we have included representative flow cytometry gating strategies, subdivided into a lineage gating and identification of naïve T cells (**new Suppl. Fig. 1a**) as well as expression of PD-1 and CD57 on CD4 and CD8 T cells (**new Suppl. Fig. 1f**). We also provide representative FACS plot for CD161, CD28+ V γ 9V δ 2 T cells and CD31+ V δ 1 T cells in the **new Figure 3g**. For examination of expressed granzymes as well as CD8A and CD8B surface expression by $\gamma\delta$ T cells, we also provide representative FACS plots (**new Fig. 4h and revised Fig. 5**).

Reviewer Figure 1.3: Representative Gating strategy for markers defining lineage. Here, we provide density plots illustrating the gating strategy used to determine cell populations in our manual gating approach. Firstly, lineage gating was performed by gating on mononuclear cells in forward (FSC-A) and side scatter (SSC-A). Next, erythrocytes were excluded by implementing a SSC-B-A vs. SSC-A gating step. Then, singlets were gated on in SSC-A vs. SSC-H, followed by gating on live cells (Zombie NIR negative). B cells were identified by their expression of CD19, while T cells were identified by their expression of CD3. Cells that did not express either CD19 or CD3 were labelled as non-B non-T cells and were later subgated (see box labelled 'non-T cells'). Subsequently, we proceeded to gate for T cells, utilising the presence or absence of the gamma-delta T cell receptor to distinguish $\gamma\delta$ and non- $\gamma\delta$ T cells. Among the TCR $\gamma\delta$ - T cells, MAIT cells were identified by their coexpression of TCRV α 7.2 and CD161. Non-MAIT T cells were then defined as $\alpha\beta$ T cells and further gated into CD4 and CD8 T cells based on CD4 and CD8 expression. Regulatory T cells were defined as CD4 T cells expressing high levels of CD25 and low levels of CD127. $\gamma\delta$ T cells were subgated based on the expression of V δ 1, V δ 2, and V γ 9, defining the two major subsets as V δ 1 T cells and V γ 9V δ 2 T cells. Non-T cells were subgated by size using FSC-A and SSC-A. Monocytes were classified into classical (CD14⁺ CD16⁻), intermediate (CD14⁺CD16⁺) and non-classical (CD14⁻CD16⁺) based on their expression of CD14 and CD16. To ensure that the non-classical monocyte subpopulation was not perturbed by CD16⁺ NK cells, pregating by size was implemented. NK cells were first gated by size and then their expression of CD16 and CD56 was determined.

The authors suggest there are no differences between the clusters in control versus CHD samples, although cluster C6 (Vd1+ cells) appears to be exclusively observed in the control sample (Figure 3B).

Answer: We appreciate the reviewer for bringing to our attention the imprecise wording regarding the cluster proportions between both groups in the manuscript. As can be seen in new **Suppl. Figure 3b**, cluster 6 is mostly, but not exclusively composed by ctrl cells. This compositional difference is reflected as well in the cluster proportions in **Fig. 3a and e**. The revised results text of the manuscript now accurately describes the cluster proportions between the two groups as follows: *“None of the clusters were specific to the children with CHD or the non-CHD control group (Fig. 3e). However, cluster c6 consisted mostly of cells from the control group.”*

The data in Figure 3C (and Supplementary 3C) suggests some level of CD4 expression by V γ 9V δ 2 T cells, which does not quite fit with what is known about these cells. Similarly, cluster 1 in Figure 3D suggests that many V γ 9V δ 2 T cells are positive for CD8 expression. The addition of conventional flow cytometric analysis would be complementary to the current data set.

Answer: We would like to thank the reviewer for this question and would like to refer to the technical details and comment on them accordingly.

The flow cytometry data have been transformed using the logicle transformation, which has been found to be advantageous for the presentation of flow cytometry data. In particular, negative or fluorescence values around 0 are difficult to display in conventional logarithmically scaled flow cytometry plots, as they are pressed against the y-axis and thus not visible to the viewer. On the other hand, high fluorescence values can be displayed with logarithmic scaling. The logicle transformation is combining the advantages of both scaling options by becoming logarithmic for large data values, to ensure a wide dynamic range and to provide good visualizations of the often log-normal distributions, at high fluorescence intensities. In reverse, the function becomes linear near zero, and extends to negative data values and is symmetrical around zero, providing near-linear visualization, appropriate for linear-normal distributions at low fluorescence intensities.

In our datasets, displaying the compartment of $\gamma\delta$ T cells in children, only about 1% of the clustered $\gamma\delta$ T cells are CD4⁺ (Reviewer Fig. 1A, B), which aligns with the publications on CD4 and CD8 expression on $\gamma\delta$ T cells from healthy adults (León-Lara et al., 2024).

Reviewer Figure 1.4. A) Flow plots and histograms showing the strategy used to manually access CD4⁺ and CD8⁺ $\gamma\delta$ T cell frequencies on pre-gated samples for live CD3⁺ TCR $\gamma\delta$ ⁺ cells used for unsupervised clustering. **B)** Dot plots showing the percentage of CD4⁺ cells and CD8⁺ within $\gamma\delta$ T cells. Statistical analyses were performed using the Mann-Whitney U test (ns $p > 0.05$, * $p < 0.05$, ** $p < 0.01$, *** $p < 0.001$, **** $p < 0.0001$). Bars indicate the median percentage per group, with each dot representing one patient. **C)** Density plots illustrating the density distribution for expression levels of surface markers per donor. Graphs are colored by group. Expression of CD4 (4th plot in the second row) is negative at a expression value of 0 to ~ 3.7 and positive at expression values above.

Due to these negligibly low positive signals for CD4⁺ γδ T cells in the dataset the transformation will tend to convert the low fluorescence intensities to linear scale values, resulting in an existing signal from still overall mostly CD4⁺ negative cells, which can be confirmed by the dot plot below and expression plots in **Reviewer Fig. 1.4A - B** and **new Supplementary Fig. 3a**. For CD4 surface expression, as can be seen in Reviewer Fig 3C, the logicle transformation transformed the negative signals as a peak between values of 0 and ~3.7. Therefore, the values displayed in the heat map, although having numeric expression values around 3 (see unscaled heat map in Reviewer Figure 2.1 and ridge plots in **new Supplementary Fig. 3d**), still reflect overall negative signal values for CD4. We have clarified this issue in the corresponding results section to read as follows: “Moreover, the eight clusters differed in the expression levels of the respective phenotypic markers (Fig. 3c, Suppl. Fig. 3d). The heat map displays scaled expression values for each marker for better comparability, highlighting marker positivity (yellow) and negativity (violet), respectively (Fig. 3d).” The exact values for when an expression value is considered a positive signal for each marker are shown in the **new Supplementary Fig. 3c**, expression plots in the Reviewer Figure below (**Reviewer Figure 1.4 C**) and by the unscaled heat map in Reviewer Figure 2.1.

Analysis of cytokine expression by Vγ9Vδ2 T cells can be achieved by short term stimulation with PMA and Ionomycin. Long term stimulation with HMBPP or anti-CD3/28 may obscure subtle differences between CHD and control groups. Granzyme B can also be analysed direct ex-vivo without the need for in vitro stimulation.

Answer: Thank you for this kind suggestion. We aimed at examining cell type specific cytokine responses and therefore chose HMBPP and anti-CD3/CD28 as suitable reagents for in-vitro stimulation of Vγ9Vδ2 T cells and CD4 and CD8 T cells, respectively. Still, we agree that the assessment on γδ T cell functionality deserves a deeper analysis.

In the revised manuscript, we generated scRNA-seq data to study the potential functionality of Vγ9Vδ2 T cells and Vδ1 T cells post-thymectomy at transcriptional level (**new Fig. 4**). Based on the scRNA-seq results, we investigated on Granzyme A and Granzyme B expression on Vγ9Vδ2 T cells and Vδ1 T cell directly ex vivo by flow cytometry (**new Fig. 5d** and **new Fig. 4h**). We now show that Vδ1 T cells have enhanced granzyme release capabilities, and that CD28⁺ Vγ9Vδ2 T cells, which produce Granzyme A show elevated frequencies in thymectomized children with CHD. We have revised the discussion, accordingly.

Discussion should focus on the origins of Vγ9Vδ2 T cells. The data provided by the authors highlight the importance of the thymus in shaping the pool of Vγ9Vδ2 T cells after birth, yet this is poorly discussed in the context of recent papers in the field. The presence of TRDJ1⁺ Vγ9Vδ2 T cells in post-natal thymus and not fetal thymus (Papadopoulou J Immunol 2019) suggests that selection in the thymus is somehow

altered after birth, rather than preferential expansion of TRDJ1+ V γ 9V δ 2 T cells in response to microbial exposure.

Answer: Thanks for pointing this out. In addition, to a potential selective expansion of TRDJ1+ T cell clones post birth, we have now included the idea of a potential selection for TRDJ3+ T cell clones in the revised discussion by stating: *It is established that the postnatal thymus produces some V γ 9V δ 2 T cells that are high for TRDJ1 element. (Perriman et al., 2023) There could be either a bias for a preferential usage of TRDJ3+ TCRs during V(D)J recombination and/or intra-thymic selection of TRDJ3+ $\gamma\delta$ T cell clones in the fetal period.*

Reviewer #2 (Remarks to the Author):

Recent work has explored the immunological consequences of thymectomies in infancy on adult immune function, and identified reduced numbers and repertoire diversity of $\alpha\beta$ T cells. However, the consequences of partial/full thymectomy on non-conventional T cell lineages, specifically those that differentiate during exclusive developmental windows, remains unknown.

Here, authors investigate the effects of partial/full thymectomy in CHD children on the development and persistence of $\gamma\delta$ T cell subsets. Authors focus on the two major thymically derived $\gamma\delta$ T cell subsets present in human blood; prenatally derived V γ 9V δ 2 T cells and postnatally derived V δ 1+ T cells. Comparison of isolated peripheral blood lymphocytes (PBMCs) between control groups and thymectomized infants with CHD, 5–12-year post-surgery, revealed reduced frequency and repertoire diversity of V δ 1 T cells and elevated frequencies of fetal-derived V γ 9V δ 2 T cells. Overall, these findings suggest that infant thymectomy results in a long-term reduction in postnatally derived V δ 1 T cells in peripheral blood.

Given the immunological importance of $\gamma\delta$ T cells in infection and anti-tumor responses, the findings of this manuscript are undoubtedly of high interest to the field, offering developmental insight into these non-conventional thymically derived T cell subsets. However, given the limitations of this study, including those mentioned by the authors at the end of the manuscript, and the overall volume, depth and complexity of the results presented, I do not think the manuscript as it currently stands contains enough novel data for publication in a journal of this impact. If authors could include more insightful functional analysis, explore the longer-term effects of this observed alter $\gamma\delta$ T cell heterogeneity in older cohorts, or delve deeper into potential alterations to additional $\gamma\delta$ T cell subsets, at both a protein and transcriptional level, this would strengthen the manuscripts conclusions greatly and would widen its impact.

Answer: We thank the Reviewer for the critical review of our manuscript. One major strength of this study is the very well defined study population of children that had a similar complexity of congenital

heart disease and received a corrective surgery with complete/partial thymectomy within the first weeks of life, where a focus on alterations on $\gamma\delta$ T cell populations was set. We agree that the phenotypic characterization and functional analysis of $\gamma\delta$ T cells deserves a more detailed investigation.

To gain deeper insights into transcriptional alterations that may underlie functionality we have generated sc-transcriptomes of $\gamma\delta$ T cells isolated from four thymectomized CHD children and four age matched controls (**presented in the new Fig. 4 – 5**), as well as longitudinally followed infants pre- and post CHD surgery (**new. Fig. 7**). Notably, we have neither the ethical vote nor patient cohort to study transcriptional profiles in longitudinally followed preterm neonates. Instead, we used two uninfected preterm neonates as the control group in the new Fig. 7, and discuss the disadvantages (specifically an enrichment of V δ 2 T cells in preterm at birth) within the revised discussion.

In summary due to the newly generated data and new experiment, we report on a reduction of naive CD31⁺ V δ 1 T cells and an increase of CD28⁺CD161⁺ (mostly) NKG2A⁺ V γ 9V δ 2 T cells in the CHD group both at transcriptional and protein level. The transcriptome analyses were further giving the basis to delve into the functional capabilities of V γ 9V δ 2 T cells and V δ 1 T cells. For V δ 1 T cells, we report that those remaining have enhanced cytotoxicity, which was measured by GZMA and GZMB production post-thymectomy. For V γ 9V δ 2 T cells we provide evidence for enrichment of CD28⁺ CD161⁺ CD8A⁺ cells that show a retained functionality with a bias towards GZMA release in the children with CHD. Together, our results highlight that early life thymectomy may enhance immune aging of postnatal-derived V δ 1 T cells in children with CHD. In contrast, early life derived $\gamma\delta$ T cell are capable to persist as long living effector cells in humans. These new results are observed at transcriptional and protein level.

As you will see in the revised manuscript version (Fig. 4- 5) we identified a specific V δ 2⁺ cluster c6, which was very abundant in the children with CHD. Those cells did not only express CD28 and CD8A, but also genes associated with innate cytotoxic $\gamma\delta$ T cells (PLZF), CCR6 and RORC, as well as tissue $\gamma\delta$ T cells in humans, which were not seen in the 6 months old infants post thymectomy (**new Suppl. Fig. 7e**). We hypothesize that 6 months are too short to see such alterations in the transcriptional profiles of $\gamma\delta$ T cells after thymectomy. Also, we could speculate that those or some of the V δ 2 T cells may recirculate from tissues in the older children with CHD, and have revised our discussion accordingly.

Unfortunately, we do not have access nor the ethical permission to study $\gamma\delta$ T cells in adults and elderly patients. Also, because clinical care of this vulnerable group of neonates that have a very critical and complex CHD has only being improved over the last 40 years, which basically means that the first surviving patients just reached adulthood. Our results might be important for the clinical care and

optimizing T cell based therapies in patients with CHD, known to have a higher risk for infections and cancer.

As studied on $\gamma\delta$ T cells are rare in the context of CHD disease (Mancebo et al., 2008), and the results of this work are very timely with recent reports on human $\gamma\delta$ T cell adaptation in early life (Gray et al., 2024; León-Lara et al., 2024), we believe that our work is now of valuable interest for reader of Nat. Communications.

Focused points:

The manuscript uses flow cytometry and TCR sequencing approaches to examine the differences in $\gamma\delta$ T cells in PBMCs from CHD patients and controls. In addition, authors perform in-vitro stimulation assays to explore functional alterations between $V\gamma9V\delta2$ T cells from the two groups. Overall, the experiments displayed seem well executed, the results are clearly displayed and presented to a high standard. However, in many figures the data displayed could be explored in greater depth, representative flow plots are absent, vital control group comparisons are missing and only specific $\gamma\delta$ T cell subsets are examined, all at a frequency level.

Answer: We appreciate these comments. As outlined above we now provide new transcriptional and functional data that give a deeper insight into $\gamma\delta$ T cell biology post-thymectomy (**new Fig. 4 -5, new Fig. 7**).

Moreover, as also suggested by the Reviewer 1 we now provide representative gating strategy for many of the experiments performed (**revised Suppl. Fig. 1, Fig.3 – 5**). As clinical data for blood counts is not available, we can only provide changes of immune cell subsets at frequency levels. Thus, and as pointed out in the answer to Reviewer 1, we analyzed the abundance of major T cell populations among lymphocytes, which gave similar results as reported frequencies within leukocytes (**revised Fig. 1**). We believe that the here reported frequencies still represent alterations within the immune cell/lymphocyte compartment.

Figure 1: Figure 1 A to H confirm known changes in $\alpha\beta$ T cells post thymectomy. The biological significance of many of the populations explored is absent from the text, including TEMRA populations and increased PD1 on T cells. No representative flow plots are displayed. Figure 1G onwards explores $\gamma\delta$ T cell subsets by flow cytometry. Representative flow plots should be displayed and Figure 1L could be presented as a ratio to give more clarity on the proportional change of $V\gamma9V\delta2$ vs all other $\gamma\delta$ T cell subsets.

Answer: Thank you for these advices. In the revised manuscript, we have added representative flow charts to identify $\gamma\delta$ T cells as well as naïve, CD31⁺, PD-1⁺ and CD57⁺ CD4 T cells within the **new Suppl. Figures 1a and 1f**. As the manuscript mainly focuses on $\gamma\delta$ T cells, we directly place findings on CD4

and CD8 T cells, namely an increase of memory T cells, CD57⁺ T cells, and lower TCR diversity post-thymectomy in the context of previous work within the results section.

For $\gamma\delta$ T cell subsets, we now included a stacked bar plot in the **new Fig. 1i** that shows proportional changes within the $\gamma\delta$ T cell compartment. The data is supported by the **Suppl. Fig. 1j**.

Figure 2: A and B: The data suggested a higher frequency of TRGV9 but a reduced Simpson's clonality score, suggesting reduced diversity. Could this be a result of clonal expansion of the increased V γ 9V δ 2 T cell population? I also think that means conclusions like "Taken together, $\gamma\delta$ TCR repertoire analyses depict a reduction of clonal diversity, which is restricted to the V δ 1 T cell subset" are inaccurate, as differences in the diversity of the TRGV9 are clear displayed.

Answer: We agree with the reviewer's point on a reduced TCR repertoire diversity within TRGV9⁺ sequences. We think that the reduced diversity is a result of a higher abundance of TRGV9⁺ sequences with low or no nucleotide insertions, which were described to be derived from the early fetal thymus by the group of David Vermijlen (Papadopoulou et al., 2019). We believe that this idea is in line with the enrichment of TRDJ3⁺ sequences in the CHD children post-thymectomy. As TRDJ3 gene usage is a very clear read-out to assess fetal-derived TCR repertoire, we focused our analyses on these data. Accordingly, we have also revised the conclusions of our results as follows: *"Together, the increase of $\gamma\delta$ T cells expressing a V γ 9V δ 2⁺ TCR with enrichment for early fetal thymus-derived TRDJ3⁺ T cell clones suggests that a reduced postnatal thymic activity supports the persistence of $\gamma\delta$ T cell clones that developed prior surgery."*

Could authors explore proliferation rates, either by flow with Ki67, and/or at a transcriptional level, comparing expression of cell cycle genes, to speculate why this change might be occurring.

Answer: The point is well taken. To specifically investigate in proliferation rate right after thymectomy and also in childhood, we have performed scRNA-seq of $\gamma\delta$ T cells isolated (i) from children with CHD and age-matched healthy controls (**new Table 2**) and (ii) from two longitudinally followed neonates pre- and post surgery of CHD (clinical data described in **revised Suppl. Table 2**). By comparing gene module scores for proliferating T cells among controls and infant with CHD (**new Fig. 7**), we observed that the V δ 1⁺ and V δ 2⁺ clusters increased proliferation rates early after thymectomy, with the largest difference for the V δ 1 T cell clusters. We indeed conclude that there is a homeostatic proliferation of $\gamma\delta$ T cells in the infants with reduced thymic activity. Notably, these differences in gene module score expression of genes for proliferation were less evident in older children with and without CHD (**new Suppl. Fig. 4**), which either suggest a saturation in the proliferation and/or (partial) recovery of thymic activity in the older children with CHD.

C and E: The data included in the treemaps from two representative donors appear to display differences in many V-gene elements and J-gene elements between the ctrl and CHD groups. However,

the intricacies of these changes are hard to identify as the data is currently displayed. Additional graphs (for example pie graphs) could explore this in more depth and the inclusion of all samples would strengthen findings across all individuals.

Answer: Thanks for pointing out. In the **revised Figure 2 and Suppl. Fig. 2**, as well as in the **revised Figure 6 and Suppl. Fig. 6** we included pie charts displaying V-J-element pairing on the group level (Main Figures 3 and 6) and at sample level (Suppl. Figure 3 and 6), which overall provides a more detailed information than the treemaps of the first manuscript version.

G: No control graph is displayed. Do authors see the same correlation in control groups with the natural variation in %RTE between individual's samples?

Answer: We appreciate this comment, and have now included respective controls in the **revised Figure 2d** (for *TRDJ3* element usage) and **revised Suppl. Fig. 2d** (for *TRDJ1* gene element usage). In brief, we do not see the same correlation in the control group, indicating that the negative correlation of *TRDJ3* usage (%) with CD31+ naïve CD4 T cells (RTE) is not explainable by natural variation but rather thymectomy as distinctive characteristic of the control group. The *TRDJ3* usage (%) in controls does not seem to be related to the number of RTE, but rather tend to reach a steady state at around 20% *TRDJ3* usage.

Figure 3: Authors use a UMAP dimension reduction approach to explore in more depth flow cytometric differences displayed in figure one. Clusters have been labelled V δ 1 or V δ 9, based on the expression of markers on a scaled heat map (figure 3C). Does the heat map display the same clear trends if unscaled?

Answer: Thanks to the reviewer for raising this question. When displaying unscaled logicle transformed values, as presented below (Reviewer Fig. 2.A), the heat map is showing the same marker characteristics of the clusters. For better comparability and visualization by standardizing each marker expression from 0 to 1, we have provided the scaled heat map in the **Fig 3c**. Nevertheless, ridge plots in **main Figure 3d** and **Suppl. Fig. 3c** also display median and full expression distributions per surface marker in an unscaled way, important to judge on the expression levels of surface markers, respectively.

Reviewer Figure 2.A. Heat map presenting unscaled median expression values for each marker and cluster c1 to c8.

Including feature plots for the most important proteins, such as V δ 1, V δ 9 and V δ 2 would add confidence to the clusters analysed.

Answer: Thank you for this suggestion. Please find feature plots for V δ 1, V δ 9 and V δ 2 expression included in new **Figure 3b** and new **Supplementary Figure 3a**, which was a further basis to annotate the clusters identified in the clustering approach of the FACS data.

Additionally, a bar graphs for the frequency of Ctl and CHD cells per cluster would help confirm changes between clusters. This would add to the analysis in figure 3E, where the frequency of all $\gamma\delta$ T cell is broken down per cluster.

Answer: Thank you for this suggestion. We now included a bar plot providing information on the proportion of cells from ctrl and CHD samples per cluster (**new Fig. 3a**).

The histograms of each marker in 3D do not allow exploration of co-expression of the surface markers. Displaying flow plots of the key markers against each other, such as CD31 vs CD8, would enable dual expression to be explored further.

Answer: Many thanks for this incentive. We now included scatter plots displaying the cluster's key markers against each other in the **new Supplementary Figure 3e**.

Figure 3E identifies phenotypical differences of V δ 1 T cells between the two groups, with lower CD31hi (with and without CD8) frequencies in the CHD group. Authors propose CD31 as a potential marker for recent thymic V δ 1 T cell emigrants, which is reduced in the CHD group. However, no data is displayed to support CD31 as a RTE marker on human $\gamma\delta$ T cells.

Answer: Thank you for highlighting the uncertainty surrounding the use of CD31 as a suitable marker for monitoring recently emigrated V δ 1 T cells. The newly generated scRNA-seq data also showed *PCEAM1* (encoding CD31) to be expressed on V δ 1 T cells, and gene module scores for RTE were decreased in CHD children as compared to controls (**new Fig. 4d-e**). Still, we think that these findings are very timely as naïve V δ 1 T cells expressing genes related to RTE were recently reported and highlighted in 6 – 8 months old preterm neonates (Gray and Farber, 2024; León-Lara et al., 2024). We agree that additional validation in larger patient cohorts and by measuring TREC levels in combination with CD31⁺ V δ 1 T cells would be required to define CD31 as a definitive marker for recently generated V δ 1 T cells. We included this idea in the revised discussion.

Figure 4:

No in-vitro functional assessment of the V δ 1 T cells is performed, although this population displays the biggest difference between control and CHD groups.

Answer: We thank the reviewer for this valuable objection and have also acknowledged this weakness before. Low frequency of V δ 1 T cells in the blood and the limited availability of sample material in this study, were complicating in vitro stimulation of V δ 1 T cells.

To now assess functional capabilities of V δ 1 T cells in children with CHD post-thymectomy and healthy children, we first conducted a scRNA-seq experiment (4 donors each) that are presented in the **new Fig. 4** and **new Fig. 5**, including **Supplements**. Overall, the scRNA-seq results confirmed the by FACS reported reduction of CD31⁺ V δ 1 T cells post-thymectomy, and an increase of V γ 9V δ 2 T cells being CD161⁺ CD28⁺ (mostly) NKG2A⁺.

To elucidate on functionality, gene module scores for cytotoxicity were calculated per cluster and per sample group (CHD vs controls). These analyses revealed a higher functionality for the V δ 1 T cell cluster, and defined V δ 2 T cell clusters. Additional FACS analysis of GZMA and GZMB production of V δ 1 T cell directly ex vivo gives evidence that V δ 1 T cells, despite their overall reduction, show enhanced functionality post-thymectomy in children (**new Fig. 4h**). Similarly, and as outlined by the answer to reviewer 1, the new FACS data shows an increase GZMA⁺ CD28⁺ CD8⁺ V γ 9V δ 2 T cells in children with CHD as compared to controls (**new Fig. 5c**). We have revised the results and discussion accordingly.

Representative flow plots from the control groups are also missing for the V γ 9V δ 2 T cells (Figure 4A).

Answer: We now show representative FACS plots of in vitro stimulation experiments for control and CHD samples within the **revised Fig. 5f-g**.

Reviewer #3 (Remarks to the Author):

Cramer et al investigated the impact of neonatal thymectomy in patients with congenital heart disease (CHD) on the persistence of $\gamma\delta$ T cells. The authors employed both flow cytometry and TCR repertoire analysis to determine how $\gamma\delta$ T cell subsets were impacted. They report that while postnatally produced V δ 1 $\gamma\delta$ T cells were diminished in thymectomized patients, fetal-derived V γ 9V δ 2, and specifically the CD161hiNKG2A⁺ subset showed elevated abundance. They also suggest that the study provides insight into how loss of the thymus is associated with changes in persistence fetally-derived $\gamma\delta$ T cells. The experiments are well designed and executed; however, there are a couple points that diminish enthusiasm.

First, the data do not support the assertion that V γ 9V δ 2 cells are increased in abundance. Panel 1K clearly shows that when normalized to leukocytes, the V γ 9V δ 2 representation is unchanged, despite there increased representation among $\gamma\delta$ cells.

Answer: We thank the reviewer for the comments and incertions. Here, an important concern was raised, which we addressed by adapting the results text accordingly: “Interestingly, V δ 1 T cells, but not V γ 9V δ 2 T cells, showed reduced frequencies within lymphocytes and total T cells in CHD children (**Fig. 1J-K, Suppl. Fig. 1G**). Along that line, the frequency of prenatal-derived V γ 9V δ 2 T cells is elevated within total $\gamma\delta$ T cells in the pediatric CHD group (**Fig. 1L**).” We admit that V γ 9V δ 2 T cells are the only T cell subset not being reduced when examining the proportion of total lymphocytes, while increase of V γ 9V δ 2 T cells is to be noticed within the $\gamma\delta$ T cell compartment. However, seeing a reduction of CD4,

CD8 and V δ 1 T cells, but not of V γ 9V δ 2 T cells within lymphocytes (Figure 1c, h) highlights that V γ 9V δ 2 T cells are affected differently by thymectomy.

Second, while the analysis is very careful, the results are somewhat expected in that V γ 9V δ 2 cells that are primarily fetally-derived persist after thymectomy, while postnatally produced V δ 1 cells are diminished as would be predicted.

Answer: Thank you for your critical review of our data. While we agree that our results align well with previously published work on $\gamma\delta$ T cells and acknowledge that some of the results are not entirely unexpected, we still believe that our work substantially contributes and advances the current state of research. Our study was conducted in a cohort of particularly vulnerable individuals, namely pediatric patients, and is therefore directly applicable and relevant to human immunology. We think that an improved understanding of the effects of thymic tissue removal in humans is of great importance, as these children face a life, fortunately with increasing life expectancy, with an altered immune system. $\gamma\delta$ T cells, described to have a higher abundance in children as in adults, have been studied rare in the context of thymectomy and/or CHD disease (Mancebo et al., 2008). We appreciate that the analysis was very descriptive in the initially submitted manuscript. As part of the revision, we have now further expanded on our results by conducting new analyses and new experiments including single cell sequencing and flow cytometric assessment of GZMA and GZMB both on V γ 9V δ 2 and V δ 1 T cells – please see the reply to reviewer 1 and 2. above We think that the new data is very timely to recently published works (Gray et al., 2024; León-Lara et al., 2024) and highlights on the biology of human $\gamma\delta$ T cells in relation to CHD disease and reduced thymic activity infancy and childhood.

References

- Ao, Y.-Q., Jiang, J.-H., Gao, J., Wang, H.-K., Ding, J.-Y., 2022. Recent thymic emigrants as the bridge between thymoma and autoimmune diseases. *Biochim. Biophys. Acta - Rev. Cancer* 1877, 188730.
- Drozdov, D., Petermann, K., Dougoud, S., Oberholzer, S., Held, L., GÜngör, T., Hauri-Hohl, M., 2022. Dynamics of recent thymic emigrants in pediatric recipients of allogeneic hematopoietic stem cell transplantation. *Bone Marrow Transplant.* 57, 620–626.
- Gray, J.I., Caron, D.P., Wells, S.B., Guyer, R., Szabo, P., Rainbow, D., Ergen, C., Rybkina, K., Bradley, M.C., Matsumoto, R., Pethe, K., Kubota, M., Teichmann, S., Jones, J., Yosef, N., Atkinson, M., Brusko, M., Brusko, T.M., Connors, T.J., Sims, P.A., Farber, D.L., 2024. Human $\gamma\delta$ T cells in diverse tissues exhibit site-specific maturation dynamics across the life span. *Sci. Immunol.* 9, eadn3954.

- Gray, J.I., Farber, D.L., 2024. $\gamma\delta$ T cells: The first line of defense for neonates. *J. Exp. Med.* 221.
- Gudmundsdottir, J., Óskarsdóttir, S., Skogberg, G., Lindgren, S., Lundberg, V., Berglund, M., Lundell, A.-C., Berggren, H., Fasth, A., Telemo, E., Ekwall, O., 2016. Early thymectomy leads to premature immunologic ageing: An 18-year follow-up. *J. Allergy Clin. Immunol.* 138, 1439-1443.e10.
- Kooshesh, K.A., Foy, B.H., Sykes, D.B., Gustafsson, K., Scadden, D.T., 2023. Health Consequences of Thymus Removal in Adults. *N. Engl. J. Med.* 389, 406–417.
- León-Lara, X., Fichtner, A.S., Willers, M., Yang, T., Schaper, K., Riemann, L., Schöning, J., Harms, A., Almeida, V., Schimrock, A., Janssen, A., Ospina-Quintero, L., von Kaisenberg, C., Förster, R., Eberl, M., Richter, M.F., Pirr, S., Viemann, D., Ravens, S., 2024. $\gamma\delta$ T cell profiling in a cohort of preterm infants reveals elevated frequencies of CD83+ $\gamma\delta$ T cells in sepsis. *J. Exp. Med.* 221.
- Mancebo, E., Clemente, J., Sanchez, J., Ruiz-Contreras, J., De Pablos, P., Cortezon, S., Romo, E., Paz-Artal, E., Allende, L.M., 2008. Longitudinal analysis of immune function in the first 3 years of life in thymectomized neonates during cardiac surgery. *Clin. Exp. Immunol.* 154, 375–383.
- Papadopoulou, M., Tieppo, P., McGovern, N., Gosselin, F., Chan, J.K.Y., Goetgeluk, G., Dauby, N., Cogan, A., Donner, C., Ginhoux, F., Vandekerckhove, B., Vermijlen, D., 2019. TCR Sequencing Reveals the Distinct Development of Fetal and Adult Human V γ 9V δ 2 T Cells. *J. Immunol.* 203, 1468–1479.
- Perriman, L., Tavakolinia, N., Jalali, S., Li, S., Hickey, P.F., Amann-Zalcenstein, D., Ho, W.W.H., Baldwin, T.M., Piers, A.T., Konstantinov, I.E., Anderson, J., Stanley, E.G., Licciardi, P. V, Kannourakis, G., Naik, S.H., Koay, H.-F., Mackay, L.K., Berzins, S.P., Pellicci, D.G., 2023. A three-stage developmental pathway for human V γ 9V δ 2 T cells within the postnatal thymus. *Sci. Immunol.* 8, eabo4365.
- van den Broek, T., Borghans, J.A.M., van Wijk, F., 2018. The full spectrum of human naive T cells. *Nat. Rev. Immunol.* 18, 363–373.

REVIEWERS' COMMENTS

Reviewer #1 (Remarks to the Author):

The revised manuscript by Ravens and colleagues is significantly improved and the data presented robust. The authors have added a significant amount of additional data that further support their initial findings to address many of the reviewer's comments. This includes the addition of single cell RNA-sequencing data sets, which is not a trivial exercise. The change in the gd TCR repertoire in control versus CHD donors highlights the importance of the post-natal thymus in shaping the T cell pool and furthermore suggests that microbial antigens are somehow influencing the selection or expansion of the Vd2+ population.

Minor comments.

- The new text contains several errors and more attention to detail is required here (lines 239, 443 and 486).
- The authors need to ensure that cited references are correct for their discussion points.
- The data trends in Figure 3H.

Reviewer #2 (Remarks to the Author):

Authors have made significant amendments to the manuscript to satisfy my major concerns. I believe the work presented is novel and impactful to the field, and therefore should be accepted for publication in Nature Communications.

Reviewer #3 (Remarks to the Author):

Cramer et al investigated the impact of neonatal thymectomy in patients with congenital heart disease (CHD) on the persistence of $\gamma\delta$ T cells using flow cytometry, TCR repertoire, scRNA-Seq analysis. While the findings of the initial submission were in many ways predictable, limiting the impact and novelty of the study, the new data that has been generated and presented in the revised manuscript substantially enhances its impact. Consequently, my view is that the study is now suitable for publication in Nature Communications.

Point-by-point reply

REVIEWERS' COMMENTS

Reviewer #1 (Remarks to the Author):

The revised manuscript by Ravens and colleagues is significantly improved and the data presented robust. The authors have added a significant amount of additional data that further support their initial findings to address many of the reviewer's comments. This includes the addition of single cell RNA-sequencing data sets, which is not a trivial exercise. The change in the gd TCR repertoire in control versus CHD donors highlights the importance of the post-natal thymus in shaping the T cell pool and furthermore suggests that microbial antigens are somehow influencing the selection or expansion of the Vd2+ population.

Minor comments.

- The new text contains several errors and more attention to detail is required here (lines 239, 443 and 486).

Answer: Thanks for the comment. We have corrected the grammar throughout the manuscript, as also indicated in the track-change modus.

- The authors need to ensure that cited references are correct for their discussion points.

Answer: We have revised the cited references in the discussion. Specifically, we have included two more citations, namely Fichtner et al. JLB 2020 to state on TRDJ3 gene element usage in neonatal and adult blood gamma delta T cells and Sumaria et al. Nature Immunology, 2024, which described CD8A and CD8B positive gamma delta T cells in mouse and humans.

- The data trends in Figure 3H.

Answer: We thank the reviewer for pointing this out. In the revised results text, we now state significant and non-significant as following: *Consistent with these analyses, a non-significant increase in CD28⁺ CD161⁺ γδ T cells and a significant decrease in CD31⁺ Vδ1⁺ cells among γδ T cells was observed in children with CHD as compared to the control group (Fig. 3g-h).*